# Discovery of exolytic heparinases and their catalytic mechanism and potential application

Qingdong Zhang[1,6], Hai-Yan Cao[2,3,4,6], Lin Wei[1], Danrong Lu[1], Min Du[1], Min Yuan[1], Deling Shi[1], Xiangxue Chen[5], Peng Wang[3,4], Xiu-Lan Chen[2,4], Lianli Chi[1], Yu-Zhong Zhang[3,4✉] & Fuchuan Li [1✉]

Heparinases (Hepases) are critical tools for the studies of highly heterogeneous heparin (HP)/heparan sulfate (HS). However, exolytic heparinases urgently needed for the sequencing of HP/HS chains remain undiscovered. Herein, a type of exolytic heparinases (exoHepases) is identified from the genomes of different bacteria. These exoHepases share almost no homology with known Hepases and prefer to digest HP rather than HS chains by sequentially releasing unsaturated disaccharides from their reducing ends. The structural study of an exoHepase (BlexoHep) shows that an N-terminal conserved DUF4962 superfamily domain is essential to the enzyme activities of these exoHepases, which is involved in the formation of a unique L-shaped catalytic cavity controlling the sequential digestion of substrates through electrostatic interactions. Further, several HP octasaccharides have been preliminarily sequenced by using BlexoHep. Overall, this study fills the research gap of exoHepases and provides urgently needed tools for the structural and functional studies of HP/HS chains.

[1] National Glycoengineering Research Center and Shandong Key Laboratory of Carbohydrate Chemistry and Glycobiology, Shandong University, Qingdao, China. [2] State Key Laboratory of Microbial Technology, Shandong University, Qingdao, China. [3] College of Marine Life Sciences, and Frontiers Science Center for Deep Ocean Multispheres and Earth System, Ocean University of China, Qingdao, China. [4] Laboratory for Marine Biology and Biotechnology, Pilot National Laboratory for Marine Science and Technology (Qingdao), Qingdao, China. [5] Tiandong Pharma, Dongying, China. [6]These authors contributed equally: Qingdong Zhang, Hai-Yan Cao. ✉email: zhangyz@sdu.edu.cn; fuchuanli@sdu.edu.cn

Heparin (HP)/heparan sulfate (HS), the polyanionic and heterogeneous polysaccharides belonging to glycosaminoglycan family[1,2], are ubiquitously presented on cell surfaces and in the extracellular matrix (ECM) as well as the intracellular environment (in mast cells)[2,3]. The various structural characteristics of HP/HS permit these molecules to interact with various proteins, such as enzyme inhibitors[3], chemokines[4], growth factors[5], morphogens[6], and other signaling proteins, to participate in various physiological and pathological processes, such as coagulation[7], cell adhesion[8], inflammation[4], cell migration[3], differentiation[9], and even pathogenic infections[10]. These vital biological roles have attracted great attention to the structural and functional studies and the clinical applications of HP/HS. Since being discovered in 1916, HP and its low-molecular preparations, as the most important class of anticoagulants, have been widely applied in clinical treatment[11].

The backbone of HP/HS polysaccharides is composed of repeating disaccharide units consisting of D-glucuronic acid (GlcA)/L-iduronate acid (IdoA) and N-acetyl-D-glucosamine (GlcNAc)[12]. During the biosynthesis process of HP/HS, their common precursors composed of repeating disaccharide GlcA–GlcNAc units are initially synthesized and then further modified by several enzymes, including N-deacetylase-N-sulfotransferase and 3-O-and 6-O-sulfotransferase, which modify GlcNAc residues; C5-epimerase, which converts the GlcA to an IdoA residue; and 2-O-sulfotransferase, which catalyzes the sulfation of GlcA/IdoA residues[13]. These modifications can result in numerous disaccharide variations in HP/HS polysaccharides, which make HP/HS the most sophisticated polymers in nature[13]. Consequently, the high structural complexity has severely hindered the structural and functional studies of HP/HS.

Heparinases (Hepases) from bacteria are polysaccharide lyases that specifically catalyze the β-eliminative reaction of glycosidic bonds between GlcNAc and GlcA/IdoA residues in HP/HS chains to produce oligosaccharides containing a 4,5-unsaturated uronic acid residue at the nonreducing end and are indispensable tools for the structural and functional studies of HP/HS[14,15]. To date, based on their substrate specificity, the identified Hepases can be divided into three types: Hepase I (EC4.2.2.7), which specifically degrades highly sulfated and IdoA-rich HP; Hepase III (EC 4.2.2.8), which prefers to degrade low sulfated and GlcA-rich HS; and Hepase II (EC 4.2.2.-), which digests both HP and HS[16,17]. Structurally, Hepase I folds into a β-jelly roll-type structure and prefers to cleave the α-1,4-linkages connected to IdoA2S/GlcA2S residues[18,19]. In contrast, Hepase III is composed of an N-terminal $(\alpha/\alpha)_5$ barrel domain and a C-terminal antiparallel β-sandwich domain and prefers to cleave the α-1,4-linkages connected to GlcA/IdoA residues[18,20]. Hepase II adopts a topology similar to Hepase III and has no selectivity for certain GlcA/IdoA or IdoA2S/GlcA2S structures[18,21,22]. All Hepases share similar catalytic mechanisms in which His-Tyr acts as a Brønsted base and acid[19,22,23]. Notably, all identified Hepases belong to the family of endolytic lyases, which randomly cleave the HP/HS chains on the inside, while exo-type Hepases, which can be very useful tools for sequencing of HP/HS chains, have not been found until now.

Here, we discover a type of Hepase with exolytic activity. These exolytic Hepases belong to the PL15 family and prefer to degrade highly sulfated HP chains at their reducing end by sequentially releasing unsaturated disaccharides. Further structural studies have shown that the active center of the exolytic Hepase is an L-shaped semiopen tunnel with a positively charged entrance and a negatively charged exit, which precisely controls the sequential cleavage of highly negatively charged HP. This exolytic mechanism is significantly different from those of other identified exolyases in the PL15 family.

## Results

**Sequence information of the Hepases.** The *biexohep* gene (GenBank: EDV07780.1) is 2607 bp long, contains 46.11% GC, and encodes a protein consisting of 868 amino acid residues. The molecular mass and isoelectric point (pI) of the putative BIexoHep protein are 98.9 kDa and 6.54, respectively. SignalP 4.1 and LipoP 1.0 analyses indicate that the BIexoHep protein has a type I signal peptide containing 23 amino acid residues. Furthermore, Carbohydrate-Active Enzyme database and Simple Modular Architecture Research Tool analyses showed that the BIexoHep protein is composed of three domains: an N-terminal signal peptide, a DUF4962 superfamily domain and a C-terminal Hepase II/III superfamily domain. Phylogenetic analysis showed that BIexoHep clustered with other putative Hepases in the PL15 family, such as the Hepase II (BTexoHep) from *Bacteroides thetaiotaomicron* VPI-5482[24] and heparan-sulfate lyase (BCexoHep) from *Bacteroides cellulosilyticus* WH2 in the PL15_2 family[25]. This subfamily of the PL15 family is separated from other Hepase clades, as shown in Supplementary Fig. 1. Notably, the enzymatic characteristics of these putative Hepases in this subfamily, including BCexoHep and BTexoHep, remain to be studied.

To investigate the possibility of BIexoHep belonging to an unidentified Hepase family, several proteins sharing various similarities with BIexoHep in the PL15_2 family, such as the Hepase II/III-like protein (BFexoHep) (GenBank: EEX44367.1) from *Bacteroides finegoldii* DSM 17565, the hypothetical protein JCM10512_3474 (BRexoHep) (GenBank: GAE85076.1) from *Bacteroides reticulotermitis* JCM 10512s, the DUF4962 domain-containing protein (PAexoHep) (NCBI accession number: WP_026063245.1) from *Pedobacter arcticus*, the DUF4962 domain-containing protein (BXexoHep) (GenBank: WP_008020419.1) from *Bacteroides xylanisolvens* and the DUF4962 domain-containing protein (PHexoHep) (GenBank: WP_015808643.1) from *Pedobacter heparinus* were synthesized for the following studies. The sequence properties of these proteins are listed in Table 1. These PL15_2 proteins were mostly composed of an N-terminal signal peptide, a DUF4962 superfamily domain and a C-terminal Hepase II/III superfamily domain, except PAexoHep and BFexoHep, which do not have the N-terminal signal peptide and the C-terminal Hepase II/III superfamily domain. In the phylogenetic tree, these proteins all clustered with BIexoHep in the PL15 family (Supplementary Fig. 1).

**Enzymatic characteristics of the Hepases.** BIexoHep was able to slowly degrade HP to mainly produce disaccharides with absorbance at 232 nm (Supplementary Fig. 2a), hardly degrade HS (Supplementary Fig. 2b). Interestingly, compared to its action on the untreated HP polysaccharides, BIexoHep showed stronger activity to the substrates prepared from the exhaustive digestion of HP polysaccharides with Hepase III (HP-F$_{\alpha III}$) (Table 1). These results indicate that BIexoHep is an HP lyase instead of an HS lyase.

To further characterize the proteins sharing various similarities with BIexoHep in the PL15_2 family, these proteins were heterologously expressed in *Escherichia coli* (E. coli) and used to estimate the enzymatic activity. The results showed that the proteins BCexoHep, BTexoHep, BFexoHep, and PAexoHep had similar substrate specificity to BIexoHep (Supplementary Fig. 2a, b). In contrast, the proteins BRexoHep, BXexoHep, and PHexoHep showed no activity against any tested polysaccharides. These results indicate that the PL15_2 family is an assembly of Hepases that prefer to degrade HP rather than HS.

The optimum temperatures of the recombined enzymes BIexoHep, BCexoHep, BTexoHep, PAexoHep, and BFexoHep are 30 °C, 20 °C, 20 °C, 10 °C, and 20 °C, respectively (Table 1). Notably, the low optimum temperature of PAexoHep may be due

**Table 1 Comparison of sequence and enzymatic properties of various PL15_2 family Hepases.**

| | | BIexoHep | BCexoHep | BTexoHep | PAexoHep | BFexoHep |
|---|---|---|---|---|---|---|
| Sequence character | Molecular mass (KDa) | 98.9 | 98.5 | 100.3 | 98.3 | 100.2 |
| | Isoelectric point | 6.54 | 6.91 | 7.37 | 8.83 | 9.76 |
| | Signal | SPI | SPII | SPII | SPI | SPI |
| Optimal condition | Temperature(°C) | 30 | 20 | 20 | 10 | 20 |
| | pH | 6.0 (HAc-NaAc) | 8.0 (Tris-HCl) | 8.0 ($NaH_2PO_4$-$Na_2HPO_4$) | 8.0 ($NaH_2PO_4$-$Na_2HPO_4$) | 8.0 ($NaH_2PO_4$-$Na_2HPO_4$) |
| | Salt | – | NaCl (230%) (100 mM) | NaCl (788%) (250 mM) | NaCl (140%) (5 mM) | KCl (485%) (50 mM) |
| Effect of divalent metal cation (5 mM) | Enhancer | $Ca^{2+}$ [a] (142%) $Ba^{2+}$ (115%) | $Ca^{2+}$ (135%) $Ba^{2+}$ (150%) $Mg^{2+}$ (133%) | $Ca^{2+}$ (192%) $Ba^{2+}$ (198%) $Mg^{2+}$ (187%) | $Ba^{2+}$ (130%) $Mg^{2+}$ (142%) | $Ca^{2+}$ (200%) $Ba^{2+}$ (262%) $Mg^{2+}$ (179%) |
| | Inhibitor | $Co^{2+}$ (0%) $Hg^{2+}$ (0%) $Pb^{2+}$ (0%) $Ni^{2+}$ (0%) $Cu^{2+}$ (0%) $Zn^{2+}$ (0%) | $Co^{2+}$ (33%) $Hg^{2+}$ (4%) $Pb^{2+}$ (28%) $Ni^{2+}$ (4%) $Cu^{2+}$ (1%) $Zn^{2+}$ (11%) | $Co^{2+}$ (36%) $Hg^{2+}$ (22%) $Ni^{2+}$ (1%) $Cu^{2+}$ (10%) $Zn^{2+}$ (10%) | $Ni^{2+}$ (0%) $Cu^{2+}$ (0%) $Zn^{2+}$ (11%) | $Co^{2+}$ (1%) $Hg^{2+}$ (31%) $Ni^{2+}$ (1%) $Cu^{2+}$ (8%) $Zn^{2+}$ (5%) |
| EDTA effect (5 mM) | | 19% | 99% | 95% | 78% | 97% |
| EGTA effect (5 mM) | | 62% | 69% | 59% | 35% | 84% |
| Enzyme activity (U/mg) | HP-$F_{\alpha III}$ | [b]49.89 ± 3.18 | 61.86 ± 2.85 | 26.75 ± 1.77 | 24.25 ± 1.24 | 12.15 ± 1.48 |
| | HP | 0.75 ± 0.06 | 1.14 ± 0.03 | – | – | – |

[a]The residual activities were shown as the percentage of that (100%) of WT-BIexoHep.
[b]The enzyme activities are the means ± S.D. for at least three experiments. Source data are provided as a Source data file.

to the cold living environment of *P. arcticus*[26]. The optimal pH of the recombined enzymes BIexoHep, BCexoHep, BTexoHep, PAexoHep, and BFexoHep were determined to be 6.0 in 50 mM NaAc-HAc buffer, 8.0 in 50 mM Tris-HCl, 8.0 in 50 mM $NaH_2PO_4$–$Na_2HPO_4$, 8.0 in 50 mM $NaH_2PO_4$–$Na_2HPO_4$ and 8.0 in 50 mM $NaH_2PO_4$–$Na_2HPO_4$, respectively (Table 1). In addition, the activities of these enzymes were significantly stimulated by 5 mM divalent cations such as $Ca^{2+}$ and $Ba^{2+}$ as well as $Mg^{2+}$ in most cases (Table 1). Moreover, the activities of the PL15_2 enzymes were strongly inhibited by divalent metal ion chelating agents, in particular EGTA (Table 1), indicating that the divalent cations may involve in the catalytic mechanism of these types of Hepases. The activities of these Hepases is strongly inhibited by various heavy metal ions, such as $Co^{2+}$, $Hg^{2+}$, $Ni^{2+}$, $Pb^{2+}$, $Cu^{2+}$, and $Zn^{2+}$ (Table 1). Moreover, salts (NaCl or KCl) at certain concentrations are strong stimulators for the above-mentioned enzymes except BIexoHep, and the activities of BCexoHep, BTexoHep, PAexoHep and BFexoHep can be optimally enhanced by 100 mM NaCl (230%), 250 mM NaCl (788%), 5 mM NaCl (140%) and 50 mM KCl (485%), respectively (Table 1). This promotion of enzyme activity by salts maybe own to the presence of salt ions enhances the binding of substrates to enzyme by removing the water coat from HP chains[27], meanwhile, salts also can affect the stability, solubility, and surface charge distribution of the enzymes to affect the activity.

Finally, the enzymatic activities of these Hepases were determined under their respective optimum reaction conditions by using HP, HS, and HP-$F_{\alpha III}$ as substrates. As shown in Table 1, the enzymatic activities of BIexoHep, BCexoHep, BTexoHep, PAexoHep, and BFexoHep toward HP-$F_{\alpha III}$ were 49.89 U/mg protein (pH 6.0, 50 mM NaAc-HAc containing 5 mM $Ca^{2+}$, 30 °C), 61.86 U/mg protein (pH 8.0, 50 mM Tris-HCl containing 5 mM $Ba^{2+}$ and 100 mM NaCl, 20 °C), 26.75 U/mg protein (pH 8.0, 50 mM $NaH_2PO_4$–$Na_2HPO_4$ containing 5 mM $Ba^{2+}$ and 250 mM NaCl, 20 °C), 24.25 U/mg protein (pH 8.0, 50 mM $NaH_2PO_4$–$Na_2HPO_4$ containing 5 mM $Mg^{2+}$ and 5 mM NaCl, 10 °C) and 12.15 U/mg protein (pH 8.0, 50 mM

$NaH_2PO_4$–$Na_2HPO_4$ containing 5 mM $Ba^{2+}$ and 50 mM KCl, 20 °C), respectively; the activities of BIexoHep and BCexoHep toward HP polysaccharides were 0.75 and 1.14 U/mg protein, respectively (Table 1). However, the enzymatic activities of BTexoHep, PAexoHep, and BFexoHep toward HP polysaccharide were too low to be accurately measured. As expected, these PL15_2 enzymes showed extremely weak activities toward HS polysaccharides.

**Digestion pattern of substrate by the PL15_2 Hepases.** To determine the action pattern of the Hepases in the PL15_2 family, the HP polysaccharides were used as substrate for digestion by each enzyme in a time-course assay as described under "Methods". As results shown in Supplementary Fig. 4a–e, all the tested PL15_2 Hepases generated only disaccharides with specific absorbance at 232 nm, and no obvious amount of the larger oligosaccharide products were detected throughout the degradation process (Supplementary Fig. 4a–e), indicating that these enzymes maybe exolytic Hepases. However, the enzyme activities of these Hepases against HP polysaccharides are too weak to determine the action pattern of these enzymes exactly.

To confirm the possible exolytic character of the PL15_2 Hepases, a saturated HP DP13 fraction without specific absorbance at 232 nm was prepared from the unsaturated Hepase III-resistant HP UDP14 fraction by treatment with $O_3$[28] and used as substrate for digestion by each enzyme in a time-course assay as described under "Methods". The results showed that all enzymes could quickly degrade HP DP13, and just like the case against HP polysaccharides the digestion of HP DP13 produced only disaccharides with specific absorbance at 232 nm during the whole process of degradation (Fig. 1a–e), confirming that these Hepases should be exolytic enzymes and cleave HP chains from the reducing end to sequentially release unsaturated disaccharides.

To further confirm the substrate-degrading direction of the PL15_2 Hepases, the HP DP13 was labeled with 2-AB and then treated with these enzymes, respectively. As the results shown in

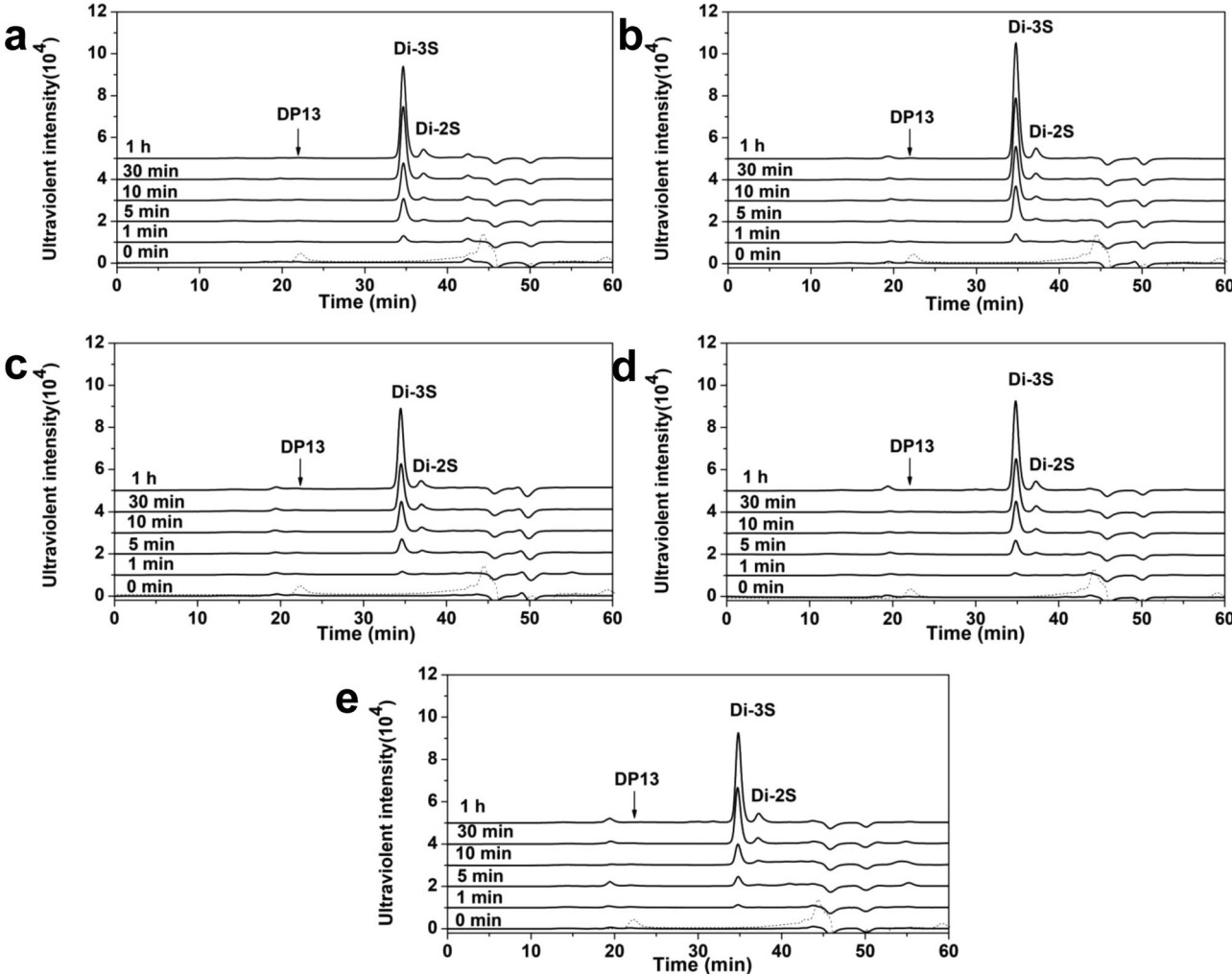

**Fig. 1 Time course experiments of degradation of saturated HP DP13 by PL15_2 family enzymes.** Saturated HP DP13 (30 μg) was treated with 50 mU of BIexoHep (**a**), BCexoHep (**b**), BTexoHep (**c**), PAexoHep (**d**) or BFexoHep (**e**). Aliquots (3 μg) were taken at different time points for SEC analysis as described under "Methods". Saturated HP DP13 was prepared via the treatment of the Hepase III-resistant HP UDP14 fraction with O₃ as described under "Methods". The elution profile of HP DP13 was detected under 210 nm and shown as dash lines in the figures. Di-3S, the trisulfated HP disaccharide; Di-2S, the disulfated HP disaccharide. The elution positions of the disaccharides were determined by comparison with those of HP standard oligosaccharides (Supplementary Fig. 3).

Supplementary Fig. 5, the 2-AB-labeling at the reducing ends of HP DP13 completely inhibited the action of the PL15_2 Hepases compared with the quick degradation of unlabeled HP DP13 by these enzymes (Fig. 1a–e), indicating that the introduction of 2-AB group at the reducing end of HP chain hindered the action of the PL15_2 Hepases and further confirmed that these enzymes are exolytic Hepases that act from the reducing ends of substrates.

**Substrate specificity of BIexoHep**. To investigate the substrate specificity of the PL15_2 Hepases, the capacity of the typical enzyme BIexoHep degrading HP tetrasaccharides with specific structures was estimated. Five structure-defined HP tetrasaccharide fractions P4-4 (ΔUA1-4GlcNAc6S1-4GlcA1-4GlcNS6S and ΔUA1-4GlcNAc6S1-4IdoA1-4GlcNS6S), P4-5 (ΔUA1-4GlcNS1-4GlcA1-4GlcNS6S and ΔUA1-4GlcNS1-4IdoA1-4GlcNS6S), P4-6 (ΔUA1-4GlcNS6S1-4GlcA1-4GlcNS6S), P4-7 (ΔUA2S1-4GlcNS6S1-4GlcA1-4GlcNS6S), and P4-8 (ΔUA2S1-4GlcNS6S1-4IdoA2S1-4GlcNS6S) were prepared and structurally determined (Supplementary Figs. 6–8 and Supplementary Table 1) and individually used as substrates for digestion assays by BIexoHep. When these tetrasaccharides (5 pmol) were

treated with excess BIexoHep overnight, all could be almost completely degraded (Fig. 2a–e), indicating that BIexoHep could digest both GlcA- and IdoA-containing tetrasaccharides regardless of their sulfation patterns. However, the tetrasaccharides (5 pmol) were treated with limited enzyme (10 mU) for 12 h, and the conversion efficiencies of tetrasaccharides P4-4, P4-5, P4-6, P4-7 and P4-8 to disaccharides were 15.90%, 9.37%, 34.68%, 100%, and 100%, respectively (Supplementary Table 2). A further study shows that the enzyme activity of BIexoHep toward P4-4, P4-5, P4-6, P4-7, and P4-8 are <1, 3.98, 24.22, 49.12, and 79.49 U/mg proteins (Supplementary Table 3), respectively, suggesting that the activity of BIexoHep against these tetrasaccharides is significantly different and that this enzyme prefers to digest highly sulfated substrates. In addition, a time course experiment showed that when size-defined HP tetrasaccharides were treated with BIexoHep the signal of Anomeric ¹H corresponding to IdoA2S residues much more quickly decreased than those of unsulfated IdoA/GlcA residues, and further nonsulfated GlcA showed more susceptible to BIexoHep than IdoA (Supplementary Fig. 9 and Supplementary Table 4). Taken together, these results suggest that highly sulfated domains in particular those containing IdoA2S residues in HP/HS chains can be the optimal substrates to exoHepases in the PL15_2 family.

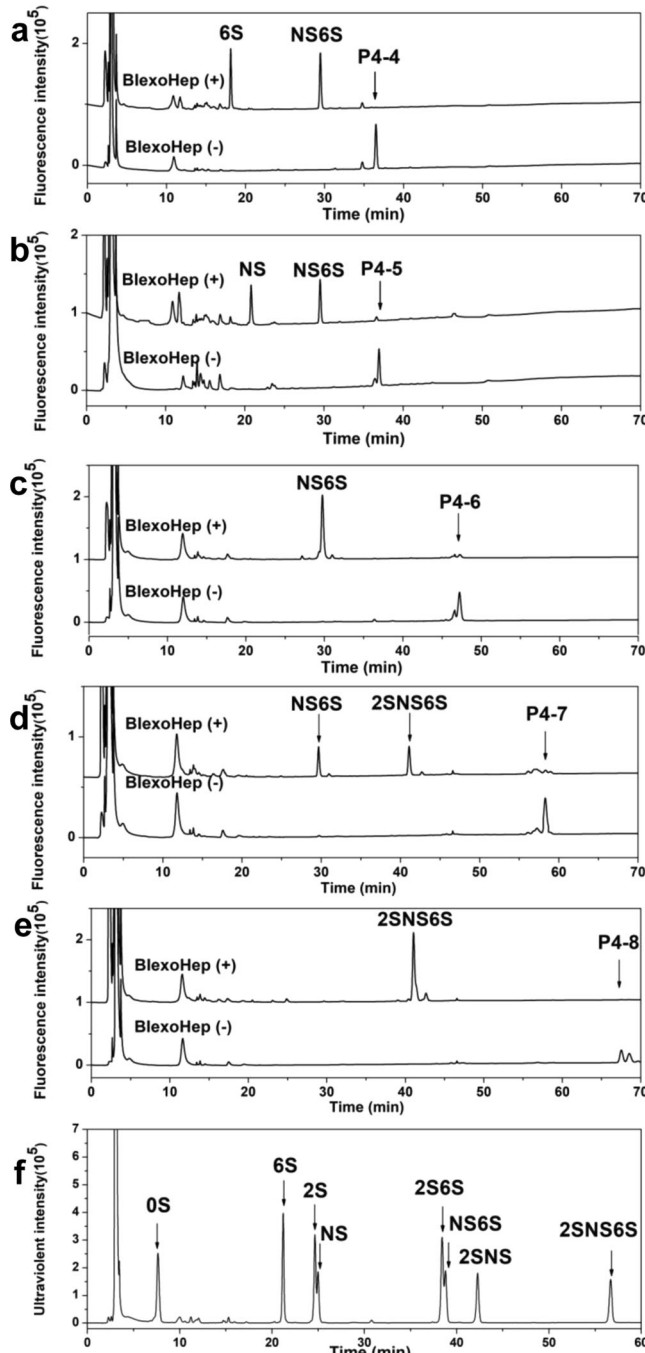

**Fig. 2 Substrate specificity of BIexoHep.** Structure-defined HP tetrasaccharides P4-4 (**a**), P4-5 (**b**), P4-6 (**c**), P4-7 (**d**), and P4-8 (**e**) were individually digested with BIexoHep, and the digests were labeled with 2-AB followed by analysis using anion-exchange HPLC on a Pack Polyamine II column eluted using a $NaH_2PO_4$ gradient (0–1 M) and monitored using a fluorescence detector with excitation and emission wavelengths of 330 and 420 nm, respectively. The types of disaccharides were determined by comparing their elution positions with those of authentic 2-AB-derivatized unsaturated HP/HS disaccharides indicated by arrows in (**f**). 0S, ΔUA(1-4)GlcNAc; 6S, ΔUA(1-4)GlcNAc6S; 2S, ΔUA2S(1-4)GlcNAc; NS, ΔUA(1-4)GlcNS; 2S6S, ΔUA2S(1-4)GlcNAc6S; NS6S, ΔUA(1-4)GlcNS6S; 2SNS, ΔUA2S(1-4)GlcNS; 2SNS6S, ΔUA2S(1-4)GlcNS6S.

As we known, Hepase I, II, and III cannot effectively work on the 3-*O*-sulfated HP substrates. To test whether the exoHepases can degrade this kind of substrates, a synthetic HP pentasaccharide, fondaparinux (Arixtra), was used as substrate to investigate the possibility. Results show that all these exoHepases can degrade the fondaparinux to generate only the unsaturated trisulfated disaccharide ΔUA2S(1-4)GlcNS6S(OCH₃) without the trisulfated HP disaccharide ΔUA(1-4)GlcNS3S6S, which is same as the case of fondaparinux treated with Hepases I[29] (Supplementary Fig. 10). These results indicate that the exoHepases could not degrade the 3-*O*-sulfated substrates, too.

**Overall structure of BIexoHep**. To investigate the structural characteristics of the PL15_2 proteins, the crystal structure of BIexoHep with an HP disaccharide product was solved at a resolution of 1.98 Å by preparing a crystal of selenomethionine (SeMet)-labeled BIexoHep in the presence of HP disaccharide products followed by single-wavelength anomalous dispersion (SAD) phasing. Furthermore, a crystal of an inactive BIexoHep-Y390A/H555A mutant with an unsaturated HP tetrasaccharide substrate was obtained, and the structure was solved at a resolution of 1.73 Å by molecular replacement with Phaser using the complex structure of BIexoHep with the product as a template. In the crystal structures, each asymmetric unit consists of one BIexoHep monomer, an unsaturated disaccharide (ΔUA2S1-4GlcNS6S) or tetrasaccharide (ΔUA1-4GlcNS6S1-4IdoA2S1-4GlcNS6S), and two $Ca^{2+}$ ions that were confirmed by inductively coupled plasma-mass spectrometry (ICP-MS) analysis. Notably, the BIexoHep complex with the disaccharide product and the structure with the tetrasaccharide substrate do not show significant differences (the root-mean-square deviation (r.m.s.d.) between the two structures was 0.140 Å based on 786 atoms).

The structure of BIexoHep consists of three domains: an N-terminal small β-sheet domain ($Ala^{24}–Asn^{150}$) composed of a large loop and a five-stranded β-sheet (β1–β5), a central $(α/α)_5$ barrel domain ($Pro^{151}–Glu^{515}$) containing 20 α-helices (α1–α20) to form an $(α/α)_5$ incomplete toroid structure, and a C-terminal β-sandwich domain ($Leu^{516}–Pro^{864}$) containing 23 β-strands and 5 short α-helices that fold into a three-layered β-sheet sandwich structure (α21–α25 and β6–β28) (Fig. 3a). The two $Ca^{2+}$ ions, named Ca1 and Ca2 are coordinated by nitrogen and oxygen atoms of several residues in near pentacoordination with distances of 3.1–3.5 Å (Fig. 3b). Ca1 is close to the uronic acid residue of the disaccharide at a distance of 5.6 Å compared with the 14 Å distance of Ca2.

A structure-based homology search for BIexoHep was performed using the DALI server[30]. The results showed that the closest structure to BIexoHep is the exo-type alginate lyase Atu3025 (PDB code 3AFL)[31] with a Z-score of 31.7 and an r.m.s. d. of 3.1 Å on 766 aligned Cα atoms. Additionally, BIexoHep shows some degree of structural similarity to Hepase II (PDB code 2FUQ, Z-score: 29.3, and r.m.s.d.: 4.3 Å on 747 atoms) from *Pedobacter heprinus* DSM 2366[21], and Hepase III (PDB code 4FNV, Z-score: 23.8, and r.m.s.d.: 4.0 Å on 659 atoms) from *P. heprinus* DSM 2366[20]. However, BIexoHep has an additional small β-sheet domain ($Ala^{24}-Asn^{150}$) at the N-terminus compared with Hepase II and III (Fig. 3c).

**The unique L-shaped catalytic cavity of BIexoHep.** Compared to the open catalytic cavities of Hepase I, II, and III (Supplementary Fig. 11a–c), BIexoHep possesses a semiopen, narrow, elongated, and L-shaped catalytic cavity composed of residues from the three domains of the N-terminal small β-sheet, central $(α/α)_5$ barrel and C-terminal β-sandwich (Supplementary Fig. 11d

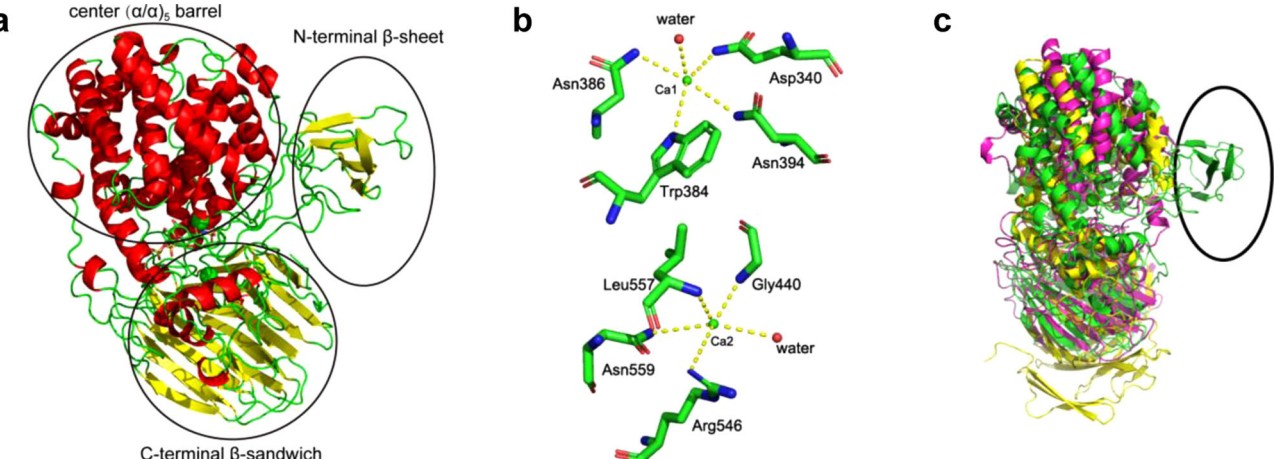

**Fig. 3 Overall structure of BlexoHep. a** Overall structure (stereo diagram) of BlexoHep. α-Helices and β-sheets are colored red and yellow, respectively. The three domains are labeled with elliptical coils. **b** Ca$^{2+}$ binding sites. The pentacoordination of Ca1 and Ca2 ions in the BlexoHep structure. A cut off distance of 3.5 Å was applied to choose neighboring residues. The distances between the residues and Ca1 atom are as follows: NE2$^{Asp340}$, 3.2 Å; NE1$^{Trp384}$, 3.3 Å; ND2$^{Asn386}$, 3.1 Å; ND2$^{Asn394}$, 3.3 Å; and water molecule, 3.3 Å. The distances between the residues and Ca2 atom are as follows: N$^{Gly440}$, 3.1 Å; NH1$^{Arg546}$, 3.4 Å; N$^{Leu557}$, 3.3 Å; ND2$^{Asn559}$, 3.5 Å; and water molecule, 3.2 Å. **c** Structural overlap of BlexoHep with Hepase II and Hepase III from *P. heprinus* DSM 2366. The structures are shown as cartoons, and Hepase II (PDB code 2FUT), Hepase III (PDB code 4MMH) and BlexoHep (PDB code 6LJL) are colored pink, yellow, and green, respectively. The additional *N*-terminal β-sheet domain of BlexoHep is labeled with an elliptical coil.

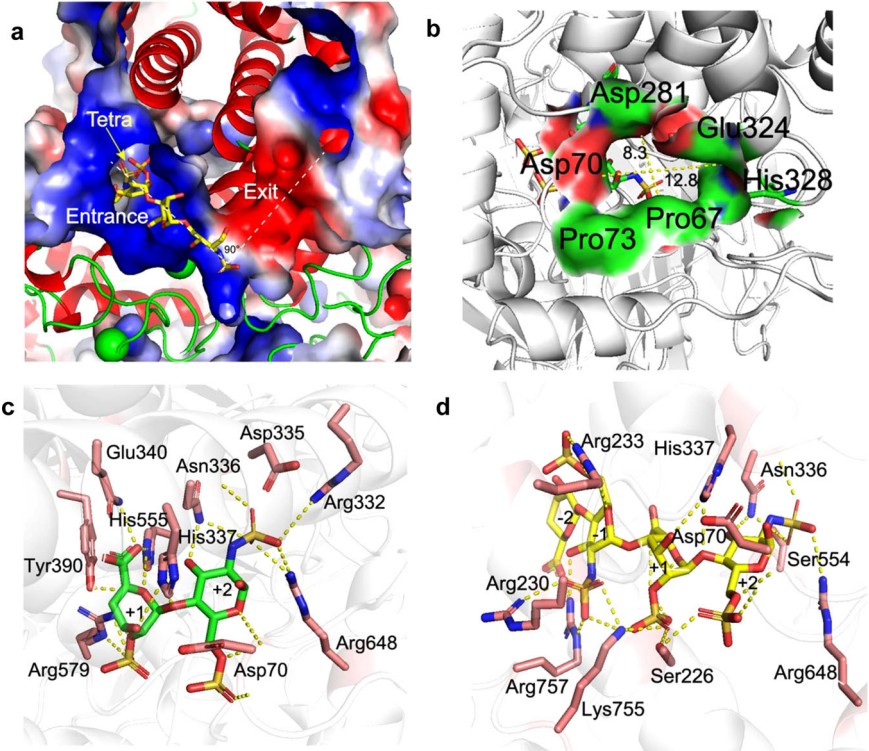

**Fig. 4 The L-shaped catalytic tunnel and relative sites of BlexoHep.** Sectional view of the L-shaped tunnel with a tetrasaccharide substrate. **a** The positively charged entrance (blue) and negatively charged exit (red) of the tunnel are indicated with white dashed lines, and the tetrasaccharide substrate is denoted by a yellow arrow. **b** Key residues in the negatively charged exit. The residues are shown in surface representation. Binding of an HP disaccharide product to the positively charged entrance of BlexoHep. **c** The disaccharide (ΔUA2S1-4GlcNS6S) is shown in thick lines and its carbon atoms are colored green, and the side chains of surrounding residues are shown in thick lines and their carbon atoms are colored fuchsia. **d** Binding of an HP tetrasaccharide substrate to the positively charged entrance of BlexoHep-Y390A/H555A. The tetrasaccharide (ΔUA1-4GlcNS6S1-4IdoA2S1-4GlcNS6S) and surrounding residue side chains are shown as yellow sticks and as fuchsia sticks, respectively. Hydrogen bonds are shown as dashed lines.

and Fig. 4a). The unique catalytic tunnel consists of two almost mutually perpendicular sections: the positively charged section generated by residues from α6, α8, α10, α12, α14, and the loops connecting α9/α10, α18/α19, β8/β9, and β10/α21, which acts as

the entrance for the binding and degradation of negatively charged HS/HP substrates, and the negatively charged section composed of residues from α7, α8, α9, and the big loop of the N-terminal small β-sheet domain, especially residues Phe$^{62}$ to Pro$^{80}$,

which acts as the exit to facilitate the release of the disaccharide products (Supplementary Fig. 12).

**Substrate-binding and product-releasing sites of the L-shaped tunnel**. According to the hydrogen-bonding networks, a series of residues, such as $Asp^{70}$, $Ser^{226}$, $Arg^{230}$, $Arg^{233}$, $Arg^{332}$, $Asn^{336}$, $His^{337}$, $Gln^{340}$, $Tyr^{390}$, $Ser^{554}$, $His^{555}$, $Arg^{579}$, $Arg^{648}$, $Lys^{755}$, and $Arg^{757}$, participate in the binding of the disaccharide product (ΔUA2S1-4GlcNS6S) and the tetrasaccharide substrate (ΔUA1-4GlcNS6S1-4IdoA2S1-4GlcNS6S) (Fig. 4c, d). The reducing end sugar rings of the tetrasaccharide substrate at the +1 and +2 subsites are tightly hugged by the side chains of the residues in the positively charged section of the L-shaped tunnel via a hydrogen bond network (Fig. 4c). The residues $His^{337}$, $Tyr^{390}$ and $His^{555}$ match well with three key residues $His^{202}$, $Tyr^{257}$ and $His^{406}$ in the active center of Hepase II from *P. heprinus* DSM 2366 (Supplementary Fig. 13), indicating that they may also play key roles in the catalytic mechanism of BIexoHep. This speculation was proven by mutation of three residues to Ala, which caused almost complete loss of enzymatic activity (Supplementary Fig. 14a, b). In contrast, the nonreducing end sugar rings at subsites −1 and −2 have fewer direct interactions with the residue sidechains because the "+" subsites are deeply buried in the tunnel, while the "−" subsites are exposed to the environment (Fig. 4d).

The negatively charged residues $Pro^{67}$, $Asp^{70}$, $Pro^{73}$, $Asp^{281}$, $Glu^{234}$, and $Asp^{335}$ from the N-terminal small β-sheet and central $(α/α)_5$ barrel domains compose the negatively charged section of the tunnel for quick delivery of the disaccharide product (Fig. 4b). Residue $Asp^{335}$ is located at the corner of the tunnel and close to the +2 subsite of the substrate (Fig. 4c), and mutation of this residue to Ala almost completely destroyed the enzymatic activity of BIexoHep (Supplementary Fig. 14a). The other residues compose the tunnel exit to approximately 8.3 Å in width (measurement of $CG^{Pro67}$ to $OD2^{Asp281}$) and 12.8 Å in length (measurement of $O^{Asp70}$ to $O^{Glu324}$). The mutation of negatively charged Asp residues to uncharged Ala or Asn or positively charged His in the exit tunnel caused a dramatic decrease of enzyme activity, such as the enzyme activities of mutants BIexoHep-P67A, BIexoHep-D70N, BIexoHep-D281A, BIexoHep-D281N and BIexoHep-D281H are reduced by 90%, 98%, 70%, 68%, and 85%, respectively, compared with that of the WT-BIexoHep (Supplementary Fig. 14a). These results show that the negatively charged residues play very important roles in the release process of disaccharide products.

**Action mode switch of BIexoHep**. To determine the key factors leading to the exolytic character of BIexoHep, the N-terminal small β-sheet domain (residues 1–150) of BIexoHep was truncated firstly, which however caused a complete loss of enzyme activity (Supplementary Fig. 14a). This finding can confirm the essential role of the additional small N-terminal domain for the activity of the exoHepases but cannot reveal the role of this domain for the exolytic property of this enzyme. Then, we tried to investigate the effect of electrical property of the exit tunnel on the action mode of BIexoHep by the mutation of the acidic residues in tunnel exit. Consistent with the observation above, the mutants BIexoHep-D70H-D281H-D335H, BIexoHep-D70H-D281N-D335H, and BIexoHep-D70N-D281N-D335N show very weak enzyme activities toward HP polysaccharides as well as the HP-$F_{αIII}$ substrates (Supplementary Fig. 14a). Interestingly, in a time-course assay, we found that the mutant BIexoHep-D70H-D281N-D335H could degrade HP polysaccharides to produce many larger size oligosaccharides during the degradation process (Supplementary Fig. 14c), which was significantly different from the action mode of the WT-BIexoHep (Supplementary Fig. 4a)

and thus showed that this mutant preliminarily possesses certain endolytic activity. These results indicate that the exolytic character of the exoHepases maybe attributed to the distribution of the negative charge in the exit tunnel but the exact reason remains to be investigated future. And the proposed exolytic mode of BIexoHep was shown in Fig. 5.

**Roles of $Ca^{2+}$ Ions**. According to the initial electron density map, there are two $Ca^{2+}$ ions Ca1 and Ca2 in the structure of BIexoHep as described above. The Ca1 is surrounded by the nitrogen and oxygen atoms of residues $Asp^{340}$, $Trp^{384}$, $Asn^{386}$, $Asn^{394}$ and a water molecule in near pentacoordination with distances of 3.1–3.3 Å (Fig. 3b). In contrast, the Ca2 is coordinated by the nitrogen and oxygen atoms of residues $Gly^{440}$, $Arg^{546}$, $Leu^{557}$, $Asn^{559}$ and a water molecule with distances of less than 3.5 Å (Fig. 3b). To investigate the roles of these two $Ca^{2+}$ ions in the catalysis of BIexoHep, the residues surrounded Ca1 and Ca2 were individually mutated to Ala and the enzyme activity of each mutant were measured. As results shown in Supplementary Fig. 14a, the activity of BIexoHep was destroyed to varying degree by the mutation. Especially, the mutation of $Asp^{340}$ surrounded Ca1 caused the enzyme to completely lose the ability to degrade HP (Supplementary Fig. 14a). These results indicate that both Ca1 and Ca2 play important roles in the catalysis of BIexoHep. In comparison, Ca1, which is much closer to substrate than Ca2, may directly participate the catalytic process of BIexoHep, while Ca2 may play an important role in the structural stabilization of enzyme. Obviously, both biochemical and structural evidence prove that BIexoHep is a $Ca^{2+}$-dependent enzyme.

**Preliminary sequencing of HP octasaccharides**. To investigating the potential application of exoHepases in the sequencing of HS/HP chains, the HP octasaccharide fraction prepared from HP-$F_{αIII}$ were subfractionated by anion-exchange HPLC on a Propack PA1 column, and five main subfractions (P8-1, P8-2, P8-3, P8-4 and P8-5) were obtained as shown in Supplementary Fig. 15. To further obtain the preliminary disaccharide sequences of these octasaccharide preparations, an enzymatic sequencing method was established based on the exolytic activity of BIexoHep (Fig. 6). Taking the sequencing of subfraction P8-4 as an example, an aliquot of P8-4 was partially digested by BIexoHep for isolating and preparing its nonreducing end hexasaccharide ($UDP6_{NE-P8-4}$) and tetrasaccharide ($UDP4_{NE-P8-4}$) fractions as described under "Methods". Then, the disaccharide compositions of P8-4 ($UDP8_{NE-P8-4}$), $UDP6_{NE-P8-4}$ and $UDP4_{NE-P8-4}$ were analyzed by complete digestion with Hepases I and II followed by 2-AB labeling and anion-exchange HPLC. As shown in Supplementary Fig. 16d, the complete digestion of $UDP8_{NE-P8-4}$ with Hepases produced two unsaturated disaccharides ΔUA1-4GlcNS6S and ΔUA2S1-4GlcNS6S with a molar ratio of 1:3, and the molar ratios of these two disaccharides in its nonreducing end hexasaccharide $UDP6_{NE-P8-4}$ and tetrasaccharide $UDP4_{NE-P8-4}$ were 1:2 and 1:1, respectively (Supplementary Table 5). By comparing the molar ratio change of disaccharides in $UDP8_{NE-P8-4}$ and $UDP6_{NE-P8-4}$, the reducing end disaccharide (D) of P8-4 could be deduced as HexUA2S1-4GlcNS6S, and similarly the disaccharide located at C site in P8-4 could be determined as HexUA2S1-4GlcNS6S by comparing the disaccharide compositions of $UDP6_{NE-P8-4}$ and $UDP4_{NE-P8-4}$ too (Fig. 6 and Supplementary Table 5). To determine the type of disaccharide at B site in P8-4, the $UDP4_{NE-P8-4}$ was treated with $O_3$ to remove the unsaturated uronic acid at the nonreducing end and further digested with Hepases I and II for disaccharide composition analysis as described in "Method". As shown in Supplementary Fig. 16d a single disaccharide peak corresponding to ΔUA2S1-

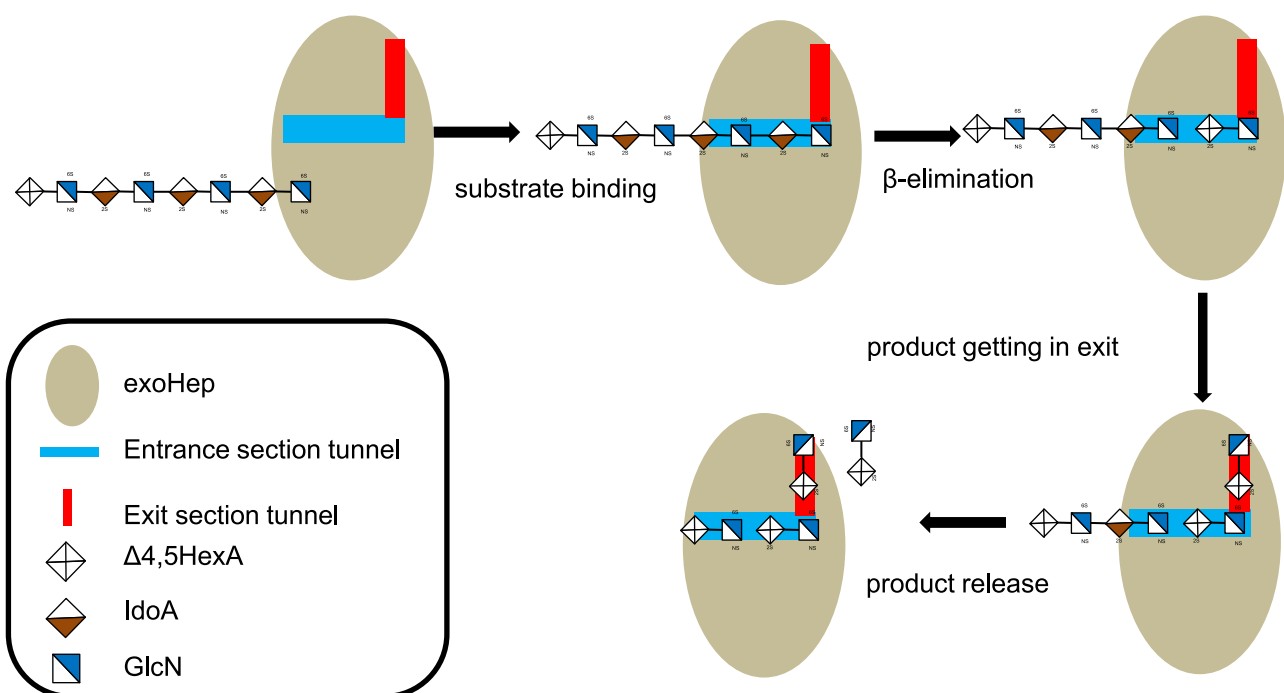

**Fig. 5 Schematic diagram of the proposed exolytic mode of BlexoHep.** The exolytic mode of BlexoHep is proposed as: first, the reducing end of the free HP chain in the environment binds to the positively charged entrance of the L-shaped tunnel via charge attraction; then the bound HP chain moves into the tunnel, and the reducing end disaccharides are cleaved via a β-elimination mechanism; finally, the resulting disaccharide product is pushed into the negatively charged exit and rapidly released.

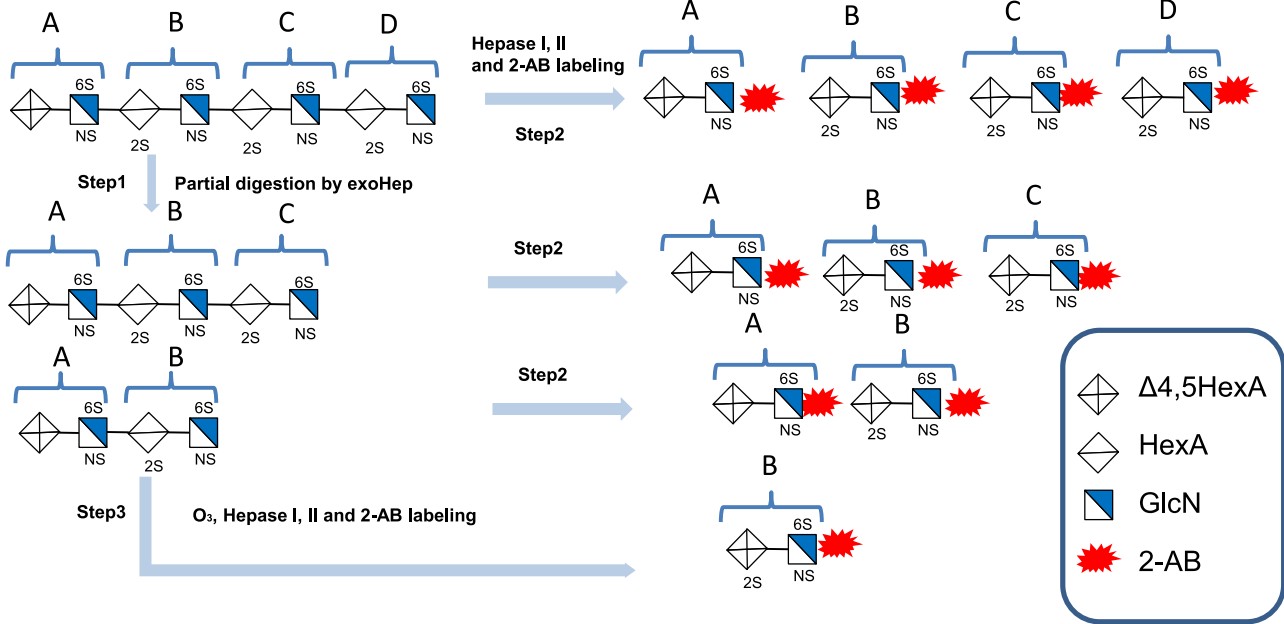

**Fig. 6 Strategy for the exo-sequencing of unsaturated HP octasaccharides.** The strategy is divided into three steps: first, the octasaccharide was partially digested by BlexoHep for isolating and preparing its nonreducing end hexasaccharide (UDP6$_{NE}$) and tetrasaccharide (UDP4$_{NE}$) fractions. Then, the disaccharide compositions of UDP8$_{NE}$, UDP6$_{NE}$ and UDP4$_{NE}$ were analyzed by complete digestion with Hepases I and II followed by 2-AB labeling and anion-exchange HPLC. By comparing the molar ratio change of disaccharides in UDP8$_{NE}$ and UDP6$_{NE}$, the reducing end disaccharide (D) could be deduced, and similarly the disaccharide located at C site could be determined by comparing the disaccharide compositions of UDP6$_{NE}$ and UDP4$_{NE}$ too. Finally, to determine the type of disaccharide at B site, the UDP4$_{NE}$ was treated with O$_3$ to remove the unsaturated uronic acid at the nonreducing end and further digested with Hepases I and II for disaccharide composition analysis. Combining all the disaccharides' information, the A site could be deduced.

4GlcNS6S was detected, indicating that the disaccharide at B site was HexUA2S1-4GlcNS6S too. Finally, based on the disaccharide composition of UDP4$_{NE-P8-4}$, the nonreducing end disaccharide (A) of P8-4 must be ΔUA1-4GlcNS6S. Taken together, the

preliminary sequence of P8-4 can be deduced as ΔUA1-4GlcNS6S1-4HexUA2S1-4GlcNS6S1-4HexUA2S1-4GlcNS6S1-4HexUA2S1-4GlcNS6S (Supplementary Table 6). By using the same method, the preliminary sequences of P8-1, P8-2, P8-3, and

P8-5 were determined as shown in Supplementary Fig. 16 and Supplementary Table 5 and Supplementary Table 6.

To verify the enzymatic sequencing, the compositions and sequences of these octasaccharides were subjected to ESI-MS (Supplementary Fig. 17 and Supplementary Table 7) and ESI-MS/MS (Supplementary Figs. 18–22). Notably, the monosulfated disaccharide located at the nonreducing end of P8-3 or P8-5 could not be deduced by the enzymatic method because of lack of it's corresponding disaccharide standard, and the exact structure of the disaccharide remains for further studies. In addition, we should note that the digestion of $UDP4_{NE-P8-3}$ produces two disaccharides ΔUA(1-4)GlcNAc(S) and ΔUA2S(1-4)GlcNS6S but the proportion of ΔUA2S(1-4)GlcNS6S is too low to hold the line with that of ΔUA(1-4)GlcNAc(S) as shown in the Supplementary Fig. 16c. The reason why the detected ratio of these two disaccharides in the nonreducing end UDP4 of P8-3 is so different is not clear, which may be due to the different labeling efficiency of the two disaccharides or the purity of P8-3.

## Discussion

In this study, the discovery of the Hepase BIexoHep with exolytic activity led to the identification of a Hepase family, PL15_2, a subfamily of the PL15 family that has been known as the assembly of exolytic alginate lyases[31], which is different from Hepases I, II, and III, belong to the PL13, PL21, and PL12 families[16], respectively. In this family, five exolytic Hepases, including BIexoHep, BCexoHep, BTexoHep, PAexoHep, and BFexoHep, have been identified and studied in detail, and compared with the previously identified Hepases (I, II, and III), all PL15_2 enzymes have an additional conserved DUF4962 superfamily domain. Structural and functional studies showed that this DUF4962 superfamily domain forms a small N-terminal β sheet domain involved in the formation of the active cleft in the 3-D structure of BIexoHep, and deletion of this domain will cause a complete loss of BIexoHep activity, indicating that the conserved DUF4962 superfamily domain is essential to the activities of PL15_2 Hepases.

Although the PL15_2 Hepases have a preference to degrade HP rather than HS, their activities against HP polysaccharides are still very low, which can be attributed to the complex structures of HP polysaccharides. The inevitable existence of some resistant structures, such as non- or low-sulfated HS domains[13] and linkage regions (GlcA-Gal-Gal-Xyl)[3,32] at the reducing ends of HP polysaccharides will prevent the exolysis of HP chains by these enzymes. This may be the key reason why this type of Hepase is not easy to find, as researchers usually use HP/HS polysaccharides as substrates to screen potential Hepases. Consistent with this hypothesis, these enzymes showed relatively higher activity against the Hepase III-resistant HP fraction HP-$F_{aIII}$. In fact, by using structure-defined HP tetrasaccharides and pentasaccharides as substrates, we found that the enzymatic activity of BIexoHep increased with the increase of sulfation degree of the substrates, except for the 3-O-sulfated substrates which resistant to the activities of the exoHepases, and further study showed that IdoA2S residues in HP tetrasaccharides were much more sensitive than nonsulfated uronic acids, which can explain why this enzyme exhibited the highest activity toward the hexasulfated tetrasaccharide ΔUA2S1-4GlcNS6S1-4IdoA2S1-4GlcNS6S. Interestingly, we also find that nonsulfated GlcA residues are more susceptible to BIexoHep than IdoA residues in tetrasaccharides, suggesting that exoHepases have different preference to Glc- and IdoA-containing saccharides. However, the detailed effects of uronic acid residues on the activity of exoHepases remain to be further investigated by using various structure-defined HP/HS oligosaccharides.

The active centers of BIexoHep and Hepase II partially overlap, indicating that their catalytic mechanism might have some features in common. As shown in this study, the residues His[337], Tyr[390], and His[555] of BIexoHep corresponding to the three key residues His[202], Tyr[257], and His[406] of Hepase II[22] play key roles in the catalytic mechanism of BIexoHep, indicating that similar to other lyases, including Hepases I, II, and III, exolytic Hepases catalyze the degradation of HP via a catalytic mechanism in which His-Tyr acts as the Brønsted base and acid[19,22,23]. However, unlike in the case of the His[337] to Ala mutation that completely destroyed the ability of BIexoHep to cleave the tetrasaccharide ΔUA2S1-4GlcNS6S1-4IdoA2S1-4GlcNS6S (Supplementary Fig. 14b), a previous study showed that the mutation of His[202] to Ala in Hepase II did not completely destroy the enzymatic activity toward this tetrasaccharide[22], indicating that there are some significant differences between the catalytic mechanisms of these two enzymes.

Compared with the open active centers of endolytic Hepase I, II, and III[19–21], BIexoHep has a semiopen active center, indicates that the substrate of HP chains cannot randomly bind to the catalytic cleft of exoHepases. Furthermore, the active center of BIexoHep is an L-shaped tunnel, which is similar to that of Hepase I from *B. thetaiotaomicron* VPI-5482[19]. However, the L-shaped tunnel of Hepase I is open, longer, and whole positively charged, which allows larger HP chains to randomly bind to the tunnel and thus results in mainly endolytic activity. In contrast, the semiopen L-shaped tunnel of BIexoHep is composed of two sections: the positively charged entrance and negatively charged exit, which can strictly control the HP chains that go through the tunnel from the positively charged entrance and be sequentially cleaved to produce an unsaturated disaccharide product that is released from the negatively charged exit. When mutate the acidic residues in the negatively charged exit to the basic amino acids, the mutant protein BIexoHep-D70H-D281N-D335H exhibited the endolytic activity. These results indicate that the exolytic character of the exoHepases might affected by the negative charge region in the semiopen L-shaped tunnel and the key reason contributes to the exolytic characters of the exoHepases are still remains for the further studies.

Based on this study, the exolytic mechanism of PL15_2 Hepases is proposed: (1) first, the reducing end of the free HP chain in the environment binds to the positively charged entrance of the L-shaped tunnel via charge attraction; (2) through electrostatic interactions, the bound HP chain moves into the tunnel, and once the reducing terminal disaccharide reaches the +1 and +2 subsites, the disaccharides will be cleaved via a β-elimination mechanism; (3) with the aid of the negatively charged residue Asp[335], the resulting disaccharide product is pushed into the negatively charged exit and rapidly released with assistance from Pro[67] and Asp[281] (Fig. 5). Notably, this exolytic mechanism is different from those of other enzymes in the PL15 family, such as the exo-type alginate lyase Atu3025[31]. The active center of Atu3025 is a pocket-like structure that releases the monosaccharide product mainly by the opening of a short α-helix, which is simple in structure and suitable for the production and release of small products. In contrast, the L-shaped tunnel of BIexoHep is more complex and suitable for the production and release of larger disaccharide products.

Currently, sequencing of HP/HS chains is still a big challenge though various techniques have been developed over the past three decades[33–35]. Compared with methods involved in mass spectroscopy and NMR, which need sophisticated instruments and professional experience, enzymatic sequencing of HP/HS saccharides is easier to be mastered and applied in common laboratories and is particularly suitable for the analysis of trace samples. However, due to the lack of exoHepases, existing enzymatic sequencing methods require a combined use of various animal-derived exoenzymes to specifically hydrolyze different

sulfate groups and glycosidic bonds, which causes the sequencing process very cumbersome[33,36]. In this study, by utilizing the exolytic feature of exoHepases that sequentially cleave HP chains from the reducing ends, an enzymatic sequencing method (Fig. 6) was established and successfully determined the preliminary sequences of five HP octasaccharides. Different from the existing enzymatic methods using specific lysosomal exoenzymes to analyze sulfate groups and monosaccharide residues one by one, this method using exoHepase can sequence HP oligosaccharides by analyzing disaccharide as a basic unit, which greatly simplifies the analysis process. Technically, this method can be used to sequence longer HP/HS oligosaccharides. Additionally, bacteria-derived exoHepases are easier to be recombinantly expressed and are more stable and higher activity comparing to animal-derived exoenzymes. We believe that these exoHepases identified in this study can be very powerful tools in the sequencing of HP/HS chains by combining with analytical techniques, such as nitrous acid degradation[33] and Ion Mobility Mass Spectrometry[35].

In conclusion, the identification of an exolytic Hepase family can provide not only enzymatic tools for structural and functional studies, particularly the sequencing of HP/HS chains, but also a clue for the discovery of additional Hepases. Moreover, revelation of the exolytic mechanism of these enzymes will certainly enrich our knowledge regarding the catalytic mechanism of exolytic lyases and be helpful for the engineering of related enzymes.

## Methods

**Materials**. The strain *Bacteroides intestinalis* DSM 17393 was purchased from German Collection of Microorganisms and Cell Cultures (Braunschweig, Germany). The genes encode PL15 family proteins were downloaded from National Center for Biotechnology Information (NCBI) database and synthesized by GENEWIZ, Inc (Suzhou, China). The Fast Mutagenesis Kit V2 was purchased from Vazyme Biotech Co., Ltd (Nanjing, China). Standard unsaturated disaccharides were purchased from Iduron (Manchester, United Kingdom). 2-Aminobenzamide (2-AB), cyanoborohydride (NaBH₃CN), HP and HS polysaccharides from porcine intestinal mucosa were provided by Tiandong Pharma (Dongying, China). All other chemicals and reagents used in this paper were of the highest quality available. Hepase I (EC 4.2.2.7), II (EC 4.2.2.-), III (EC 4.2.2.8), 2-*O*-sulfatase and glycuronidase were cloned from *Pedobacter heprinus* DSM 2366[37,38]. The anti-Hepase III fraction of HP (HP-F$_{\alpha III}$) was prepared by exhaustively degraded of HP polysaccharides with Hepase III and then precipitated with 5 volume anhydrous alcohol followed by centrifugation to collect the deposit. Size-defined HP oligosaccharide fractions were prepared by the digestion of HP polysaccharides with Hepase II or Hepase III followed by size exclusion chromatography (SEC) on a Superdex Peptide 10/300 GL column[39].

**Sequences analysis of the gene and protein of BIexoHep**. The gene and protein sequence of BIexoHep were downloaded from the NCBI database (GenBank: EDV07780.1). Then, the analysis of the similarity between BIexoHep and the known Hepases sequences was performed using the BLASTp algorithm online. Secretion signal peptides and their types were analyzed using the SignalP 4.1 server and LipoP 1.0 server online, respectively. The molecular mass of the protein was calculated using the peptide mass tool on the ExPASy server of the Swiss Institute of Bioinformatics. Sequence alignment and phylogenetic analysis were performed using MEGA version 7.0. Protein modules and domains were identified using the Simple Modular Architecture Research Tool, Pfam database, and Carbohydrate-Active Enzyme database.

**Heterologous expression and purification of BIexoHep and Its derivative with selenomethionine**. To express BIexoHep in *E. coli* strains, the gene *biexohep* was constructed onto plasmid pET-30a (+) (Novagen). Then, the plasmid pET-30a-*biexohep* was transformed into *E. coli* BL21 (DE3) cells. *E. coli* cells harboring an expression vector (pET30a-*biexohep*) were initially cultured in LB broth and then induced to start the expression of wild-type BIexoHep (WT-BIexoHep) proteins with 5 mM IPTG (isopropyl 1-thio-β-D-galactopyranosid) when the cell density reached an A$_{600}$ of 0.6–1.0. An overexpression system for the BIexoHep derivative with selenomethionine (SeMet- BIexoHep) was also constructed in *E. coli* BL21 (DE3) cells as previously reported[40]. Cells grown overnight in LB medium were harvested by centrifugation at 3700 × *g* for 5 min and washed twice with M9 medium. Then, the cells were transferred into M9 medium containing 1 mM MgSO₄, 0.1 mM CaCl₂, and 0.4% (w/v) glucose and cultured at 37 °C. When the

A$_{600}$ reached 0.6, the culture was cooled to 22 °C, and then 100 mg/L lysine, phenylalanine, and threonine and 50 mg/L isoleucine, leucine, valine, and L-SeMet were added. Fifteen minutes later, the culture was incubated at 16 °C overnight under the induction of 5 mM IPTG.

The cells were collected by centrifugation at 8000 × *g* for 5 min, washed with ice-cold buffer A (50 mM Tris-HCl, 150 mM NaCl, pH 8.0) and resuspended in the same buffer, followed by ultrasonic disruption of cells (50 repetitions, 5 s) in an ice-cold environment. After centrifugation at 15,000 × *g* for 30 min, the supernatant containing soluble proteins was collected for further steps. The supernatant was loaded onto a column packed with nickel-Sepharose 6 Fast Flow and washed with a gradient concentration of imidazole ranging from 50 to 250 mM. Then, the 250 mM imidazole eluate was diluted and loaded onto a Source 15Q column and washed with a gradient concentration of NaCl ranging from 0 to 1 M. Finally, the fractions containing the target protein were concentrated and further fractionated by gel filtration on a Superdex G-200 column. The purity of the BIexoHep protein was analyzed using SDS-PAGE. SDS-PAGE was performed using 13.2% polyacrylamide gels.

**Expression and preliminary characterization of PL15_2 genes**. A series of unidentified homogenous genes of exoHepases were synthesized by GENEWIZ, Inc. (Suzhou, China) and characterized as described for BIexoHep above. These genes, including Hepase II (BTexoHep) from strain *Bacteroides thetaiotaomicron* VPI-5482, heparan-sulfate lyase (BCexoHep) from *Bacteroides cellulosilyticus* WH2, Hepase II/III-like protein (BFexoHep) (GenBank: EEX44367.1) from strain *Bacteroides finegoldii* DSM 17565, the hypothetical protein JCM10512_3474 (BRexoHep) (GenBank: GAE85076.1) from strain *Bacteroides reticulotermitis* JCM 10512, the DUF4962 domain-containing protein (PAexoHep) (NCBI accession number: WP_026063245.1) from *Pedobacter arcticus*, the DUF4962 domain-containing protein (BXexoHep) (GenBank: WP_008020419.1) from *Bacteroides xylanisolvens* and the DUF4962 domain-containing protein (PHexoHep) (GenBank: WP_015808643.1) from *Pedobacter heparinus*, share a sequence similarity of 66.26%, 60.57%, 49.27%, 47.79%, and 31.13% with BIexoHep, respectively. The expression plasmids were transferred into *E. coli* BL21 (DE3) cells and over-expressed. The activities of the expression products were analyzed as described for BIexoHep above.

**Biochemical characterization of the recombinant PL15_2 family of proteins**. The optimal conditions for each PL15_2 protein were determined as described previously[41]. Briefly, to determine the optimal pH for each protein, HP-F$_{\alpha III}$ (1 mg/ml) was digested with the indicated protein (0.5 µg) in buffers with a gradient of pH values, including a final concentration of 50 mM NaAc-HAc buffer (pH 5.0–6.0), 50 mM NaH₂PO₄-Na₂HPO₄ buffer (pH 6.0–8.0), and 50 mM Tris-HCl buffer (pH 7.0–10.0) in a total volume of 100 µl, at 30 °C for 30 min. After the optimum pH was determined, the effects of temperature on each protein activity were tested in each optimal buffer at temperatures ranging from 0 to 70 °C for 30 min. The effects of metal ions/chelating reagents (5 mM) on the HP-F$_{\alpha III}$ degrading activities of each protein were investigated at the optimum pH and temperature determined above. The effects of salt concentrations on the activities of each protein against HP-F$_{\alpha III}$ were investigated at the optimum pH and temperature of each corresponding protein.

**Activity assay of PL15_2 enzymes**. The activities of PL15_2 enzymes were measured using HP, HS, and HP-F$_{\alpha III}$ as substrates. Briefly, each enzyme (1 µg) was added to 1 mg/ml HP, HS or HP-F$_{\alpha III}$ in its optimal buffer (50 mM) and an additional optimal concentration of ions or salts in a total volume of 1 ml. The reaction mixture was incubated at the corresponding optimal temperature. At various time intervals 0, 0.5, 1, 2, and 5 min, aliquots of 100 µl were withdrawn in duplicate, boiled for 10 min, and then cooled in ice-cold water for 10 min. After being centrifuged at 15,000 × *g* for 10 min, the supernatant was collected, and diluted three times. All reactions were performed in triplicate, and the activity of enzyme was estimated by measuring the absorbance of the diluted products at 232 nm. The activity was calculated according to the change of absorbance per minute applying a molar extinction coefficient of 3800 M⁻¹ cm⁻¹ for products [1 U = 1 µmol of (ΔUA)-containing product formed per min].

**Degradation pattern of substrates by PL15_2 enzymes**. To determine the degradation patterns of the PL15_2 enzymes, the digests of HP (1 mg/ml) by the enzymes (10 unit/ml) were traced at each optimal condition. Aliquots (20 µg) of the reactants were removed at different time points and analyzed through SEC on a Superdex Peptide 10/300 GL column eluted with 0.20 M NH₄HCO₃ at a flow rate of 0.4 ml/min and monitored at 232 nm using a UV detector. Online monitoring and data analysis (e.g., molar ration determination) were performed using the software LC solution version 1.25.

To further confirm the digest patterns and directions of the enzymes, HP UDP14, derived from the final products of HP degraded by Hepase III, was used to investigate the action pattern of the PL15_2 family of proteins. First, 200 µg of HP UDP14 was treated with O₃ to remove unsaturated uronic acid at the nonreducing ends[28], which resulted in saturated HP DP13 without specific absorbance at 232

nm. After desalting by gel filtration, saturated HP DP13 (30 μg) was digested with various PL15_2 family proteins (50 mU) and traced under their optimal conditions. Aliquots (3 μg) of the products were removed at different time points and analyzed through SEC as described above.

To further confirm the digest directions of the enzymes, the 1 μg of the HP DP13 was 2-AB labeled in advance and then treated with 5 mU of each PL15_2 enzymes for 4 h. Then the products were analyzed by SEC using a fluorescence detector with excitation and emission wavelengths of 330 and 420 nm, respectively.

**Substrate specificity of BIexoHep.** To determine the substrate specificity of BIexoHep, 5 pmol of various structure-defined HP tetrasaccharides (P4-4 (mixture of ΔUA1-4GlcNAc6S1-4GlcA1-4GlcNS6S and ΔUA1-4GlcNAc6S1-4IdoA1-4GlcNS6S), P4-5 (mixture of ΔUA1-4GlcNS1-4GlcA1-4GlcNS6S and ΔUA1-4GlcNS1-4IdoA1-4GlcNS6S), P4-6 (ΔUA1-4GlcNS6S1-4GlcA1-4GlcNS6S), P4-7 (ΔUA2S1-4GlcNS6S1-4GlcA1-4GlcNS6S), and P4-8 (ΔUA2S1-4GlcNS6S1-4IdoA2S1-4GlcNS6S)) (Supplementary Figs. 6, 7, 8 and Supplementary Table 1) were degraded by 100 mU of BIexoHep protein overnight. Then, the products were 2-AB labeled and analyzed by anion-exchange HPLC on a Pack Polyamine II column eluted using a $NaH_2PO_4$ gradient (0–1 M) and monitored using a fluorescence detector with excitation and emission wavelengths of 330 and 420 nm[42], respectively.

To determine the preference of BIexoHep to different uronic acid residues, size-defined tetrasaccharides (3 mg) prepared from the partial digest of HP treated with Hepase II was treated with 0.9 U BIexoHep for 0 min, 1 min, 5 min, 3 min or 12 h and then boiled to stop the reactions. Resulted oligosaccharides in each reactant were purified and pooled through SEC on a Superdex Peptide 10/300 GL column eluted with 0.20 M $NH_4HCO_3$, and freeze-dried repeatedly to remove $NH_4HCO_3$. Each oligosaccharide sample was dissolved in 0.5 ml of 99.9% $D_2O$ and lyophilized for twice, and was finally dissolved in 0.5 mL $D_2O$ in a 5 mm NMR tube for the following analysis. The type of the internal uronic acids in each tetrasaccharide preparation was directly determined by $^1H$ NMR spectroscopy. $^1H$ NMR spectroscopy was performed on a JNM-ECP600 (JEOL, Japan) instrument set at 600 MHz and analyzed on MestReNova (9.0.1).

To determine if the exoHepases can degrade the 3-O-sulfated HP substrates, a synthetic HP pentasaccharide, fondaparinux (Arixtra), GlcNS6S1-4GlcA1-4GlcNS3S6S1-4IdoA1-4GlcNS6S, was used as substrate to test the preference of BIexoHep and other PL15_2 enzymes. Briefly, 5 μg Arixtra was treated with each enzyme and then the product was analyzed by anion-exchange HPLC on a Pack Polyamine II column eluted using a linear gradient of $NaH_2PO_4$ from 16 mM to 450 mM and monitored at 232 nm using a UV detector.

**Mutation of BIexoHep.** To investigate the key residues involved in the active center of BIexoHep, the residues in and around the active center were individually mutated to Ala or His and Asn, using the Fast Mutagenesis Kit V2 from Vazyme Biotech Co., Ltd (Nanjing, China). The primers are listed in Supplementary Table 8. The residual activities of the mutants were measured by using HP-$F_{\alpha III}$ as substrate, and compared with that (100%) of the WT-BIexoHep. Furthermore, the activity of the mutant BIexoHep-H337A was tested by using the structure-defined tetrasaccharide subfraction P4-8 as substrate (5 pmol), and the resultants were 2-AB labeled and analyzed by anion-exchange HPLC on a Pack Polyamine II column as described above.

To investigate the action mode of the mutant BIexoHep-D70H-D281N-D335H, the degradation process of HP polysaccharide by this mutant was traced in a time-course assay as described for the WT-BIexoHep above.

**Crystallization and data collection.** The protein BIexoHep (12 mg/ml) was premixed with HP disaccharides (10 mg/ml) and then crystallized at 18 °C by the sitting drop method in buffer containing 16% (w/v) polyethylene glycol (PEG) 1500 and 100 mM MMT (DL-malic acid, MES, and Tris) (pH 6.0). The crystal of the SeMet-BIexoHep protein was prepared under the same conditions used for BIexoHep except for at pH 5.3. The inactive mutant BIexoHep-Y390A/H555A (8 mg/ml) mixed with HP UDP4 at a molar ratio of 1:10 was crystallized at 18 °C by the hanging drop method in buffer containing 100 mM MMT (DL-malic acid, MES, and Tris) (pH 6.5) and 23% PEG 1500. X-ray diffraction data were collected on a BL17U1 beam line at the Shanghai Synchrotron Radiation Facility[43]. The initial diffraction datasets were processed by HKL2000. Data collection statistics are shown in Supplementary Table 9.

**Structure determination and refinement.** The selenium single-wavelength anomalous dispersion (Se-SAD) phasing method was used to determine the phase of BIexoHep. Experimental phases were solved using the Phenix program[44] AutoSol. Initial model building was performed using the Phenix program Auto-Build. Refinement of the BIexoHep structure was performed alternately by the Phenix program Refine and Coot[45]. The ion type was confirmed by inductively coupled plasma-mass spectrometry (ICP-MS) analysis. The structure of the BIex-oHep-Y390A/H555A complex with a tetrasaccharide was determined by molecular replacement with Phaser using the complex structure of BIexoHep with a disaccharide as the search model. Data collection and refinement statistics are

summarized in Supplementary Table 9. All figures of the structures were generated using the program PyMOL.

**Preparation and sequencing of HP octasaccharides.** Size-defined HP octasaccharide fraction prepared from HP-$F_{\alpha III}$ was further subfractionated by anion-exchange HPLC on a Propac PA1 column eluted using a NaCl gradient (0.5–1.5 M) in 70 min by monitoring at 232 nm. The main peaks (P8-1, P8-2, P8-3, P8-4, and P8-5) were collected and desalted through SEC on a G10 column.

To determine the sequences of the octasaccharides P8-1, P8-2, P8-3, P8-4, and P8-5, an enzymatic sequencing method was established as shown in Fig. 6. Firstly, 500 pmol of each octasaccharides were partially digested by 50, 50, 50, 9, and 1.5 mU BIexoHep proteins for 5 min, respectively. The UDP6 and UDP4 products were collected and freeze-dried repeatedly to remove $NH_4HCO_3$ to get the purified oligosaccharides. Secondly, the UDP8, UDP6 and UDP4 of the octasaccharides were individually digested by Hepases I and II, 2-AB labeled and then analyzed by anion-exchange HPLC on a Polyamine II column eluted using a $NaH_2PO_4$ gradient (0–550 mM) in 60 min and monitored using a fluorescence detector with excitation and emission wavelengths of 330 and 420 nm, respectively. Thirdly, the nonreducing end UDP4 from the octasaccharides were individually treated with $O_3$, then digested by Hepases I and II, 2-AB labeled and analyzed as described in step 2.

The molecular mass and composition of these octasaccharides were further analyzed and confirmed by HILIC-ESI-MS on a LTQ-Orbitrap XL FT MS (Thermo Fisher Scientific, San Jose, CA) as previously reported[46]. LC separation of octasaccharides was performed on a Thermo Ultimate 3000 system (Thermo Fisher Scientific, San Jose, CA) using a Luna HILIC column (2.0 mm × 150 mm, 200 Å, Phenomenex, Torrance, CA). Mobile phase A was 5 mM ammonium acetate, while mobile B was 5 mM ammonium acetate in 98% acetonitrile. Mobile B was started at 90% and lasted for 5 min, then decreased to 65% in 20 min and kept for 5 min at a flow rate of 150 μl/min. The MS parameters included a spray voltage of −4.2 kV, a capillary voltage of −40 V, a tube lens voltage of −50 V, a capillary temperature of 275 °C, a sheath gas flow rate of 20, an aux gas flow rate of 5. All FT mass spectra were acquired at a resolution 60,000 with m/z 300–1500.

To further analyze and verify the sequences of the octasaccharides, each desalted fraction dissolved in 50% methanol containing 3 mM NaOH was directly introduced to the ESI-MS interface of a Thermo LTQ-Orbitrap XL MS (Thermo Fisher Scientific, San Jose, CA) as previously reported[47]. The MS parameters included a spray voltage of −3.5 kV, a capillary voltage of −40 V, a tube lens voltage of −50 V, a capillary temperature of 275 °C, a sheath gas flow rate of 10, an aux gas flow rate of 2. The MS/MS parameters were set as following: Iso width (m/z), 3.0; normalized collision energy, 55.0–60.0.

**Statistics and reproducibility.** Each experiment was done at least three times by triplicates. Statistical analyses were performed using Excel and Origin 8.0. Error bars represent means of triplicates ± SD. For comparison of the statistical differences between two groups, Student's t-test (two-sided) was carried out for statistical analysis.

**Reporting summary.** Further information on research design is available in the Nature Research Reporting Summary linked to this article.

## Data availability

All data supporting the findings of this study are available within the paper (and its Supplementary Information files). All relevant data generated during this study or analyzed in this published article (and its Supplementary Information files) are available from the corresponding author on reasonable request. The atomic coordinates and structure factors of the structures in this study have been deposited in the Protein Data Bank, www.pdb.org (PDB ID codes 6LJA and 6LJL). The sequences of BIexoHep (GenBank: EDV07780.1), BTexoHep (GenBank: AAO79757.1), BCexoHep (GenBank: ALJ58962.1), PAexoHep (GenBank: WP_026063245.1) and BFexoHep (GenBank: EEX44367.1) have already existed in NCBI database. The raw MS data of P8-1, P8-2, P8-3, P8-4, and P8-5 have been uploaded onto figshare and can be accessed by the link https://doi.org/10.6084/m9.figshare.13413584. Source data are provided with this paper.

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

## Acknowledgements

We thank the staffs from BL18U1&BL19U1 beamlines of National Facility for Protein Sciences Shanghai (NFPS) and Shanghai Synchrotron Radiation Facility, for assistance during data collection. We thank Xiaoju Li and Haiyan Sui from Life Science General Research Technology Platform of SKLMT (State Key Laboratory of Microbial Technology, Shandong University) for the assistance in XRD experiments. This work was supported by the National Science Foundation of China (31971201, 31570071, U1706207, 31630012, and 91851205), the Science and Technology Development Project of Shandong Province (No.2018GSF121002), the Project of Taishan Industry Leading Talent of Shandong Province (tscy20160311), the National Key R&D Program of China (2018YFC1406700, 2018YFC0310704), the Program of Shandong for Taishan Scholars (tspd20181203), AoShan Talents Cultivation Program Supported by Qingdao National Laboratory for Marine Science and Technology (2017ASTCP-OS14), and the Major Scientific and Technological Innovation Project (MSTIP) of Shandong Province (2019JZZY010817).

## Author contributions

F. Li, Y. Z. Zhang, and Q. Zhang designed research; Q. Zhang, L. Wei, D. Lu, M. Du, M. Yuan, and D. Shi performed the research; Q. Zhang, H.Y. Cao, P. Wang, and X. L. Chen, L. Wei, L. Chi analyzed data; X. Chen prepared some key substrates; Q. Zhang and F. Li wrote the paper.

## Competing interests

The authors declare no competing interests.
