## [Peer Review File · Nature Communications]

REVIEWER COMMENTS

Reviewer #1 (Remarks to the Author):

The manuscript provides a substantial contribution to the limited tool set for the analysis of heparin/HS. Currently most labs use three enzymes to degrade polymers of 20-200 disaccharides in length. Considering bacteria edit heparin / HS chains to gain entry into the cells, there is huge potential for the discovery of enzymes to sequence GAG chains which is relatively unexplored. This manuscript provides a stepping stone for that requirement, in identifying multiple heparin exoenzymes, which can be utilised in the overall goal of glycosaminoglycan sequencing to further understand the interplay of numerous biological processes. The tetrasaccharide sequencing and analysis is excellent, characterisation of the enzymes using multiple analytical methods is of a high standard, the only weakness in the manuscript is the octasaccharide sequencing, which would benefit from MS/MS or NMR data to confirm interpretation of the HPLC disaccharide analysis. Overall, this is a manuscript that would be of high interest to the heparin field.

Line 36 – please remove “a kind of” and replace with “are” heterogeneous anionic polysaccharides.

Line 37 – Please add “family” belonging to the glycosaminoglycan family

Line 45 – “since being discovered in the 1920” it was 1916, clinical 1930’s. Please correct sentence accordingly.

Line 55 – Can the authors remove the number “32” as I’m assuming this is excluding the free amines, which is perfectly acceptable, however, the theoretical disaccharide number is debated. It is also unlikely that all 32 exist, so a theoretical number should include all combinations.

Figure 1, please add a chromatogram for the starting heparin dp13 material. Please also run a disaccharides with 2 sulfates, one sulfate and 0 sulfates as a calibration curve for easier interpretation to the reader. Possibly add dp2, dp4, dp6, dp8, dp10 calibration SEC chromatogram. Please add SEC separation within the title.

Line 518 – As previously reported – there is no reference

Please describe Fig 1 further in the results section. There is a reference to Fig 1 on line 98, but little explanation. Fig 1 is nicely described on line 164, could the authors please put these points together.

Line 121 – “Resistant fractions” please remove this type of wording, as in heparin resistant structures often contain a 3O-sulfation which is not the aim of this study. Could the authors please perform an experiment with exoHep on Arixtra please, as it has the potential to digest highly sulphated structures like arixtra based on the tetrasaccharide and octasaccharide data in the manuscript.

Please also use Arixtra in a SEC experiment with BCexoHep, BTexoHep, PAexoHep and BFexoHep.

S4, Could the authors please have separation of the common 8 heparin disaccharide standards on the Pack Polyamine II column at the top of the figure.

General enquiry, could the authors comment on the chromatography performance of the ProPAC PA1 column vs the Pack Polyamide II column. This is of generally interest, and can be placed in methods, but not necessary.

Could the authors please provide an explanation on whether P8-4 could have the UA-GlcNS6S in the middle of the sequence for example, Δ 4HexUA2S1-4GlcNS6S1-UA1-4GlcNS6S1-4HexUA2S1-4GlcNS6S1-4HexUA2S1-4GlcNS6S. Perhaps MS/MS data or NMR data should be provided to complement the HPLC data.

Please upload raw Mass Spec data files for intact dp8 oligosaccharides.

S12 (c) The Dp4 shows that one disaccharide is observed, therefore must contain two GlcNAc disaccharides, in table S5 there is only 1 disaccharide. Could the authors please clarify there chromatograms further.

S12 – for each disaccharide profile, it is perhaps possible to determine the reducing and non-reducing end disaccharides in some cases, but it is unclear on how the two middle disaccharides are deduced to generate the overall sequence. MS/MS or NMR data could easily supplementary this disaccharide analysis to provide confident octasaccharide sequences.

Reviewer #2 (Remarks to the Author):

This manuscript described by Zhang et al. deals with molecular identification, structure determination, and potential application of exotype heparinases (exoHeps). Recombinant proteins classified into subgroup 2 of polysaccharide lyase family 15 (PL15_2) were identified to be heparin lyases exolytically acting on heparin oligosaccharides rather than heparin and heparan sulfate polysaccharides, although these proteins have previously been annotated to such enzymes due to the presence of heparinase II/III-like motif. Based on determination of crystal structure of the *Bacteroides intestinalis* PL15_2 exoHep, exoHeps were found to include the additional N-terminal small β -sheet domain distinct from other structurally resembled heparinases (Hep II and Hep III). This β -sheet domain contributes to formation of L-shaped active cleft probably involved in the exolytic activity. Substrate specificity of the exoHep using various heparin oligosaccharides was thoroughly investigated, indicating that this enzyme become a powerful tool for determination of structure of heparin, while physiological function of the enzyme in the bacterial cells remains to be clarified.

The work is technically sound, and advances the knowledge in this field, indicating the results obtained here are considered to be informative and valuable. On the other hand, several significant concerns are raised as follows:

Major points:

1) Exotype heparinase

The authors mentioned that all identified heparinases belong to the family of endolytic lyases (lines 74-76), while there are few reports describing the endolytic activity of heparinases. Some heparinases (e.g. the enzyme from *Bacteroides stercoris* HJ-15) have been found to produce unsaturated heparin disaccharides as main products from "heparin polysaccharides", although it is unclear whether these disaccharides are final products or not. In this work, the authors determined the mode of action (exotype) of PL15_2 enzymes by using "heparin oligosaccharides (DP13)" as a substrate, though these enzymes can act on heparin polysaccharide (lines 158-159). The authors should also show time-dependent degradation profile (similar to Figure 1) by using heparin polysaccharides as a substrate. If PL15_2 enzymes show an exotype activity, no other products except for unsaturated disaccharides will be observed at any reaction time (min and h). In Figure S2, a peak at retention time of around 18 min was observed only in the presence of enzymes. What is the peak? The authors specify the product.

2) Structural determinants for exolytic activity

The authors mentioned that the additional N-terminal small β -sheet domain is essential to exolytic activity (lines 328-329). If the truncated enzyme with a lack of the small β -sheet domain shows an "endolytic" activity but not loses the enzyme activity, this mention will be understood. Regarding exo/endotype polysaccharide lyases, similar studies have previously been reported in PL11 lyases (JBC, 284, 10181-10189, 2009). In this paper, exotype polysaccharide lyase has been converted to endotype lyase through identification of structural determinants of these lyases for mode of action. Thus, the authors should demonstrate structure-based conversion of exotype to endotype or vice

versa in this work. The readers of this Journal are interested in the mode of action of the PL13, PL21, and PL12 heparinases with an addition of small β -sheet domain.

Minor points:

- 3) Line 24: "seemingly" is in contradiction with lines 74-76.
- 4) Line 29: "is essential to the exolytic activity" should be changed to "is essential to the enzyme activity".
- 5) Line 50: "N-acetyl" should be changed to "N-acetyl". N is in an italic form.
- 6) Line 86: Specify the organism of the exohep gene. *Bacteroides intestinalis*?
- 7) Lines 128-129: What is the reason why BRexoHep, BXexoHep, and PHexoHep showed no enzyme activity?
- 8) Lines 144-148: Can the authors discuss the promotion of the enzyme activity in the presence of NaCl or KCl?
- 9) Lines 220-224: Two calcium ions are included in the exoHep and situated around the active site. On the other hand, EDTA showed little effect on the enzyme activity of exoHep (Table 1). Did the authors examine the effect of EGTA on the enzyme activity? The significance of calcium ions should be discussed.
- 10) Line 262-272: The role of "exit structure" by negatively charged residues was described, while this is not direct evidence for release of the product from "exit". The authors are encouraged to show the enzyme activity of the mutants with acidic residues (Asp and Glu) to basic and acidic (Glu and Asp) residues.
- 11) Line 271: "Fig S5a" should be changed to "Fig S10a".
- 12) Line 324: (I, II and II) to (I, II and III).
- 13) Lines 363-373: The authors mentioned that L-shaped tunnel include the entrance by positively charged residues and the exit by negatively charged residues. In relation to 10), replacement of acidic residues with basic residues in the exit structure may provide useful information on the role of the exit. The positively charged exit region of the mutants would exhibit the affinity with the product, indicating the role of the exit as pathway for release of the product.
- 14) Line 436: sequence to sequences.
- 15) Lines 511-512: The authors should indicate molar extinction coefficient.
- 16) Line 542: D2O "2" subscript.
- 17) Line 633: Remove 2003?
- 18) Lines 683-684: Insert space in "catalyticmechanism" and "crystalstructure".
- 19) Line 687: Remove dot after Protein.
- 20) Line 722: delta OK? Or Δ .

21) Figure1: What is small peak “Di”? The authors should isolate the product and determine its molecular weight.

22) Figure S2: The authors should show the time-dependent degradation profile. What is the peak at retention time of around 18 min?

Reviewer #3 (Remarks to the Author):

The authors discovered a novel Heparase with exolytic activity, exoHep. This exoHep showed high activity against heparase III resistant heparin fractions, relatively low activity against the heparin and extremely weak activities towards HS. The authors confirmed the exolytic cleavage and substrate specificity using heparase III resistant heparin and tetrasaccharide. This enzyme degrades highly sulfated HP from the reducing end. This is very unusual as nearly all enzymes acting exolytically on carbohydrates act from the non-reducing end.

The authors elucidated the protein structure and protein-heparin interactions through crystallization of SeMet labeling followed by single wavelength anomalous dispersion phasing.

Finally, the authors used this exoHep successfully sequenced the 5 octasaccharide. They proposed the mechanism of the exolytic mode of exoHep and strategy for exo-sequencing using exoHep.

This new exoHep provides a new tool for HP/HS sequencing.

The manuscript is well written.

1. The most unusual feature of these exo enzymes is that they act from the reducing end. While the author's data suggest this is the case they should apply a second method to confirm. The author's should test whether it is possible to disrupt enzyme action by first reducing the hemiacetal at the oligosaccharides reducing end or by reductively aminating the reducing end with 2-aminoacridone. This would help confirm that action does take place from the reducing end.
2. The authors should also use RI detector instead of UV 232nm used in Fig 1? This way they can see both oligosaccharide and disaccharide changing at the same time.
3. What's the purity of the five octasaccharides? Did author confirm the octasaccharide chain using another sequencing method for comparison to the exoHep sequencing method?
4. Figure 5, the color of exoHep is too light. It is suggested to use a darker background.

Our responses to the reviewers' comments have been described below one by one referring to the pages and locations of the marked manuscript in yellow.

Reviewer # 1 (the main findings of the study):

The manuscript provides a substantial contribution to the limited tool set for the analysis of heparin/HS. Currently most labs use three enzymes to degrade polymers of 20-200 disaccharides in length. Considering bacteria edit heparin / HS chains to gain entry into the cells, there is huge potential for the discovery of enzymes to sequence GAG chains which is relatively unexplored. This manuscript provides a stepping stone for that requirement, in identifying multiple heparin exoenzymes, which can be utilised in the overall goal of glycosaminoglycan sequencing to further understand the interplay of numerous biological processes. The tetrasaccharide sequencing and analysis is excellent, characterisation of the enzymes using multiple analytical methods is of a high standard, the only weakness in the manuscript is the octasaccharide sequencing, which would benefit from MS/MS or NMR data to confirm interpretation of the HPLC disaccharide analysis. Overall, this is a manuscript that would be of high interest to the heparin field.

Reviewer #1 (Revisions needed):

(Comments)

1. Line 36 – please remove “a kind of” and replace with “are” heterogeneous anionic polysaccharides.

Our response:

Thanks a lot. We have changed as suggested.

2. Line 37 – Please add “family” belonging to the glycosaminoglycan family.

Our response:

Thanks. We have added as suggested.

3. Line 45 – “since being discovered in the 1920” it was 1916, clinical 1930's. Please correct sentence accordingly.

Our response:

Thanks. We have corrected it.

4. Line 55 – Can the authors remove the number “32” as I'm assuming this is excluding

the free amines, which is perfectly acceptable, however, the theoretical disaccharide number is debated. It is also unlikely that all 32 exist, so a theoretical number should include all combinations.

Our response:

Thanks. We have removed and corrected as suggested.

5. Figure 1, please add a chromatogram for the starting heparin dp13 material. Please also run a disaccharides with 2 sulfates, one sulfate and 0 sulfates as a calibration curve for easier interpretation to the reader. Possibly add dp2, dp4, dp6, dp8, dp10 calibration SEC chromatogram. Please added SEC separation within the title.

Our response:

Thanks for your concerns. As we know, the saturated HP DP13 does not have specific absorbance at 232 nm, and thus no obvious signal corresponding to it is observed in the chromatograms. To show the presence of substrate in Fig. 1, we have showed the SEC chromatogram of the saturated HP DP13 detected at 210 nm. In addition, we have added a SEC calibration curve of size-defined HP oligosaccharides including HP UDP2 (disaccharides with 3 sulfates, 2 sulfates, one sulfate and 0 sulfate), UDP4, UDP6, UDP8, UDP10, UDP12 and UDP14 in the Supplement data (the new Supplementary Fig. S3). At the meantime, we also added SEC separation in the corresponding figures as you suggested.

6. Line 518 – As previously reported – there is no reference

Our response:

Thanks. We have added the reference.

7. Please describe Fig 1 further in the results section. There is a reference to Fig 1 on line 98, but little explanation. Fig 1 is nicely described on line 164, could the authors please put these points together.

Our response:

Thanks. We have noticed that the reference to Fig 1 in line 98 is a mistake and in fact it should be the reference to Supplementary Fig. S1 and we described it in the corresponding part. Sorry for this mistake.

8. Line 121 – “Resistant fractions” please remove this type of wording, as in heparin

resistant structures often contain a 3O-sulfation which is not the aim of this study. Could the authors please perform an experiment with exoHep on Arixtra please, as it has the potential to digest highly sulphated structures like arixtra based on the tetrasaccharide and octasaccharide data in the manuscript.

Our response:

Thanks for your suggestions. We have performed an experiment to investigate the digestion of Arixtra with exoHep (BlexoHep) in the revised manuscript. As shown in the new Supplementary Fig. S10, BlexoHep could digest the Arixtra only generate the trisulfated disaccharide $\Delta\text{UA}2\text{S}(1-4)\text{GlcNS}6\text{S}(\text{OCH}_3)$ but not the trisulfated HP disaccharide $\Delta\text{UA}(1-4)\text{GlcNS}3\text{S}6\text{S}$, same as the case of Hepases I found in a previous study (Clin Appl Thromb Hemost. 2001 Jan;7(1):58-64.). These results indicate that the 3-O-sulfation resists the digestion by BlexoHep too.

9. Please also use Arixtra in a SEC experiment with BCexoHep, BTexoHep, PAexoHep and BFlexoHep.

Our response:

Thanks for your suggestions. As shown in the new Supplementary Fig. S10, we have also added the experiments with BCexoHep, BTexoHep, PAexoHep and BFlexoHep on Arixtra. Same as the case of BlexoHep, these enzymes could digest the Arixtra to generate the trisulfated disaccharide $\Delta\text{UA}2\text{S}(1-4)\text{GlcNS}6\text{S}(\text{OCH}_3)$ only, suggesting that the action of PL15_2 enzymes are resisted by the 3-O-sulfation of HP/HS chains.

10. S4, Could the authors please have separation of the common 8 heparin disaccharide standards on the Pack Polyamine II column at the top of the figure.

Our response:

Thanks for your suggestions. We have added the separation curve of the common 8 heparin disaccharide standards on the Pack Polyamine II column at the bottom of the new figure S7 in the revised manuscript.

11. General enquiry, could the authors comment on the chromatography performance of the ProPAC PA1 column vs the Pack Polyamide II column. This is of generally interest, and can be placed in methods, but not necessary.

Our response:

Thanks for your concerns. Based on our experience, compared with the Pack Polyamine II column, the ProPAC PA1 column shows better capacity for the separation of HP/HS oligosaccharides in particular bigger oligosaccharides such as hexa- and octasaccharides. However, considering the fact that the Pack Polyamine II column is much cheaper than the ProPAC PA1 column and good enough to separate HP/HS disaccharides and tetrasaccharides, thus when we analyze small HP/HS oligosaccharides we prefer to use Pack Polyamine II column rather than ProPAC PA1 column.

12. Could the authors please provide an explanation on whether P8-4 could have the UA-GlcNS6S in the middle of the sequence for example, Δ 4HexUA2S1-4GlcNS6S1-UA1-4GlcNS6S1-4HexUA2S1-4GlcNS6S1-4HexUA2S1-4GlcNS6S. Perhaps MS/MS data or NMR data should be provided to complement the HPLC data.

Our response:

Thanks for your concerns. We have performed the MS/MS analysis of all five HP octasaccharides to complement the HPLC sequencing data of (the new supplementary Fig. S18-22). As shown in the ESI-MS/MS spectra of P8-4 (the new supplementary Fig. S21), the ions $[Y_6+8Na]^4-$, $[Z_6+7Na]^2-$ and $[C_2+Na]^2-$ all indicate that the disaccharide UA-GlcNS6S is located at the nonreducing end of P8-4, and none of the fragments can be detected to support that the UA-GlcNS6S residue is in the middle of the sequence.

13. Please upload raw Mass Spec data files for intact dp8 oligosaccharides.

Our response:

Thanks for your concern. We have upload the raw Mass Spec data files for each intact dp8 oligosaccharide to figshare and can be accessed by the link <https://figshare.com/s/3c029e519df2d0b1625d>.

14. S12 (c) The Dp4 shows that one disaccharide is observed, therefore must contain two GlcNAc disaccharides, in table S5 there is only 1 disaccharide. Could the authors please clarify there chromatograms further.

Our response:

Thanks a lot. Based on your concerns we noticed this problem. To clarify this question, we have repeated this experiment and found that the DP4 composed of two disaccharides Δ UA(1-4)GlcNAc(S) and Δ UA2S(1-4)GlcNS6S but the proportion of Δ UA2S(1-4)GlcNS6S

is too low to hold the line with that of Δ UA(1-4)GlcNAc3S as shown in the supplementary Fig. S16c of the revised manuscript. The reason why the detected ratio of these two disaccharides in the non-reducing-end DP4 of P8-3 is so different is not clear, which may be due to the different labeling efficiency of the two disaccharides or the purity of P8-3. However, both the result from the analysis of the reducing-end disaccharide of the DP4 (supplementary Fig. S16c) or the MS/MS data of P8-3 (the new supplementary Fig. S20) showed that the octasaccharide Δ 4HexUA1-4GlcNAc3S1-UA1-4GlcNS6S1-4HexUA2S1-4GlcNS6S1-4HexUA1-4GlcNS6S was the main component of P8-3.

15. S12 – for each disaccharide profile, it is perhaps possible to determine the reducing and non-reducing end disaccharides in some cases, but it is unclear on how the two middle disaccharides are deduced to generate the overall sequence. MS/MS or NMR data could easily supplement this disaccharide analysis to provide confident octasaccharide sequences.

Our response:

Thanks for your concerns. As mentioned above, we have added the MS/MS data of each sequenced octasaccharide to complement the HPLC data.

Reviewer # 2 (the main findings of the study):

This manuscript described by Zhang et al. deals with molecular identification, structure determination, and potential application of exotype heparinases (exoHeps). Recombinant proteins classified into subgroup 2 of polysaccharide lyase family 15 (PL15_2) were identified to be heparin lyases exolytically acting on heparin oligosaccharides rather than heparin and heparan sulfate polysaccharides, although these proteins have previously been annotated to such enzymes due to the presence of heparinase II/III-like motif. Based on determination of crystal structure of the *Bacteroides intestinalis* PL15_2 exoHep, exoHeps were found to include the additional N-terminal small β -sheet domain distinct from other structurally resembled heparinases (Hep II and Hep III). This β -sheet domain contributes to formation of L-shaped active cleft probably involved in the exolytic activity. Substrate specificity of the exoHep using various heparin oligosaccharides was thoroughly investigated, indicating that this enzyme become a powerful tool for determination of structure of heparin,

while physiological function of the enzyme in the bacterial cells remains to be clarified.

The work is technically sound, and advances the knowledge in this field, indicating the results obtained here are considered to be informative and valuable. On the other hand, several significant concerns are raised as follows:

Reviewer #2 (Revisions needed):

(Comments)

1. Exotype heparinase

The authors mentioned that all identified heparinases belong to the family of endolytic lyases (lines 74-76), while there are few reports describing the endolytic activity of heparinases. Some heparinases (e.g. the enzyme from *Bacteroides stercoris* HJ-15) have been found to produce unsaturated heparin disaccharides as main products from “heparin polysaccharides”, although it is unclear whether these disaccharides are final products or not. In this work, the authors determined the mode of action (exotype) of PL15_2 enzymes by using “heparin oligosaccharides (DP13)” as a substrate, though these enzymes can act on heparin polysaccharide (lines 158-159). The authors should also show time-dependent degradation profile (similar to Figure 1) by using heparin polysaccharides as a substrate. If PL15_2 enzymes show an exotype activity, no other products except for unsaturated disaccharides will be observed at any reaction time (min and h). In Figure S2, a peak at retention time of around 18 min was observed only in the presence of enzymes. What is the peak? The authors specify the product.

Our response:

Thanks for your concerns and suggestions. Regarding the heparinases from *Bacteroides stercoris* HJ-15, although their substrate-degrading patterns were not investigated in details we compared the sequence similarities of the Hepase I, Hepase II and Hepase III from *Bacteroides stercoris* HJ-15 with the typical Hepases from *Pedobacter heparinus*. As shown in the Supplementary Fig S1, the three Hepases from *Bacteroides stercoris* HJ-15 are individually clustered with the endolytic Hepase I, II and III from *Pedobacter heparinus* but far from all the exolytic PL15_2 enzymes, indicating that the three Hepases from *Bacteroides stercoris* HJ-15 are most likely the endolyases. Based on your suggestion, we

added the time-dependent degradation profiles of HP polysaccharides by the exoHepases. As shown in the new supplementary Fig. S4 of the revised manuscript, the unsaturated disaccharides were accumulated with the prolongation of degradation time but larger oligosaccharides were not obviously detected during all reaction time. Thus, all the results from the action-pattern assays of the PL15_2 family enzymes on both HP polysaccharide (the new supplementary Fig. S4) and oligosaccharide (DP13) (Fig. 1) show that these novel exoHepases belong to exolyases. The peak at retention time of around 18 min in supplementary Fig. S2 represents polysaccharide/large oligosaccharide and enzyme protein that eluted in the void volume, which is often detected in the size exclusion chromatogram, and we specified the peaks in supplementary Fig. S2 as suggested.

2. Structural determinants for exolytic activity

The authors mentioned that the additional N-terminal small β -sheet domain is essential to exolytic activity (lines 328-329). If the truncated enzyme with a lack of the small β -sheet domain shows an “endolytic” activity but not loses the enzyme activity, this mention will be understood. Regarding exo/endotype polysaccharide lyases, similar studies have previously been reported in PL11 lyases (JBC, 284, 10181-10189, 2009). In this paper, exotype polysaccharide lyase has been converted to endotype lyase through identification of structural determinants of these lyases for mode of action. Thus, the authors should demonstrate structure-based conversion of exotype to endotype or vice versa in this work. The readers of this Journal are interested in the mode of action of the PL13, PL21, and PL12 heparinases with an addition of small β -sheet domain.

Our response:

Thanks for your good suggestions. As mentioned in the manuscript, the deletion of the N-terminal small β -sheet domain will leading to the thorough inactivation of BlexoHep, meaning that the additional N-terminal small β -sheet domain is essential to the activity of BlexoHep. Additionally, we also tried to link the small β -sheet domain to the PL13, PL21, and PL12 heparinases and test the activities of the engineered enzymes. However, we found that the grafting resulted in the thorough inactivation of the PL13 (E-HepI) and PL21

(E-HepII) heparinases (figure A and B below). In contrast, the grafting mutation did not cause the inactivation of PL12 heparinases (E-HepIII) (figure B), but the action pattern of E-HepIII was not affected obviously (figure C) compared with that of the wild HepIII (figure D). These results indicate that the additional N-terminal small β -sheet domain is not the decisive factor for the exolytic activity of the PL15_2 family enzymes. While, interestingly, we obtained some other progresses in the followed mutation experiments. We attempted to mutate the acidic residues Asp in the exit tunnel to uncharged Asn or the basic residue His so that the chargeability of the exit tunnel was converted to uncharged or positively charged. As results, the mutants BlexoHep-D70H-D281H-D335H, BlexoHep-D70N-D281N-D335N and BlexoHep-D70H-D281N-D335H show the relatively low enzyme activities toward the substrates. However, we were surprised to find that one of the mutants, BlexoHep-D70H-D281N-D335H, exhibited obviously endolytic activity (the new supplementary Fig. S14c). These results indicate that the exolytic character of the exoHepases maybe attributed to the distribution of the negative charge in the exit tunnel, but the exact reason remains to be investigated future.

Figure. The activities of the engineered Hepases (*Pedobacter heparinus*). A. Degradation of HP by E-HepI and E-HepII. B. Degradation of HP by E-HepII and E-HepIII. C. Time

course experiments of degradation of HP by E-HepIII. D. Time course experiments of degradation of HP by HepIII. The samples were analyzed through SEC on a Superdex Peptide 10/300 GL column eluted with 0.20 M NH_4HCO_3 at a flow rate of 0.4 ml/min and monitored at 232 nm using a UV detector. E-HepI, the grafting mutated Hepases I; E-HepII, the grafting mutated Hepases II; E-HepIII, the grafting mutated Hepases III; Hexa, the HP hexasaccharide; Tetra, the HP tetrasaccharide; Di-2S, the disulfated HP disaccharide; Di-1S, the monosulfated HP disaccharide; Di-0S, the nonsulfated HP disaccharide.

3. Line 24: “seemingly” is in contradiction with lines 74-76.

Our response:

Thanks a lot. We have revised the corresponding sentence.

4. Line 29: “is essential to the exolytic activity” should be changed to “is essential to the enzyme activity”.

Our response:

Thanks for your suggestion. We have revised the sentence as you suggested.

5. Line 50: “N-acetyl” should be changed to “N-acetyl”. N is in an italic form.

Our response:

Thanks a lot. We have corrected it.

6. Line 86: Specify the organism of the exohep gene. *Bacteroides intestinalis*?

Our response:

Thanks for your concern. We have renamed exohep to BlexoHep in order to indicate its source (*Bacteroides intestinalis*).

7. Lines 128-129: What is the reason why BRexoHep, BXexoHep, and PHexoHep showed no enzyme activity?

Our response:

Thanks for your concern. The three proteins BRexoHep, BXexoHep, and PHexoHep possess the sequence similarities of 60.57%, 47.79% and 31.13% with BlexoHep (old name exoHep), respectively, and have the conserved active residues found in all exoHepases, but they have no activity on all tested polysaccharides. Frankly, we do not know the exact reason why these three proteins are inactive, which may be due to some difference in three dimensional structure compared with the active exoHepases. In fact,

the current method to find new enzymes or other functional proteins based on the homology comparison of primary sequences often leads to the cloning and expression of many proteins with no activity or unknown function. Thus, it is not enough to predict the activity of a protein based on the primary structure alone, and further structural and functional study is necessary.

8. Lines 144-148: Can the authors discuss the promotion of the enzyme activity in the presence of NaCl or KCl?

Our response:

Thanks for your suggestion. As discussed in a previous literature (Glycoconj J. 2019 Jun;36(3):227-236.), the salt could remove the water coat from GAGs and shields their negatively charges. Thus, salts in the reaction system could expose the negative charge of HP and then enhance the interactions of HP and enzymes. Moreover, salts also can affect the stability, solubility and surface charge distribution of the enzymes to affect the activity of the enzyme. In fact, as shown in Table 1, except for BlexoHep, the enzymatic activities of BCexoHep, BTexoHep, PAexoHep and BFeexoHep can be stimulated by the alkaline ions Li^+ , Na^+ or K^+ , and the most suitable alkaline ions for BCexoHep, BTexoHep, PAexoHep and BFeexoHep are Na^+ , Na^+ , Na^+ and K^+ , respectively. However, the reason why exoHep cannot be enhanced by the alkaline ions remains to investigate further.

9. Lines 220-224: Two calcium ions are included in the exoHep and situated around the active site. On the other hand, EDTA showed little effect on the enzyme activity of exoHep (Table 1). Did the authors examine the effect of EGTA on the enzyme activity? The significance of calcium ions should be discussed.

Our response:

Thanks for your concern. As shown in Table 1, when 5 mM EDTA presents in the reaction system, the enzyme activity of BlexoHep sharply decreases to only 19% of that of WT-BlexoHep (old name exoHep), suggesting that EDTA can strongly inhibit the enzyme activity of BlexoHep. As you suggested, we also performed the effect experiments of EGTA on the enzyme activities of the exoHepases, and as expected, EGTA could significantly inhibit not only the activity of BlexoHep but also those of others compared with EDTA, indicating that these exoHepases are Ca^{2+} -dependent enzymes.

Further, to investigate the roles of these two Ca^{2+} ions in the catalysis of BlexoHep, the residues surrounded Ca1 and Ca2 were individually mutated to Ala and the enzyme activity of each mutant were measured. As results shown in supplementary Fig. S14a, the activity of BlexoHep was destroyed to varying degree by the mutation. Especially, the mutation of Asp³⁴⁰ surrounded Ca1 caused the enzyme to completely lose the ability to degrade HP (supplementary Fig. S14a). These results indicate that both Ca1 and Ca2 play important roles in the catalysis of BlexoHep. Obviously, both biochemical and structural evidence prove that BlexoHep is a Ca^{2+} -dependent enzyme. And we have made the corresponding description in the revised manuscript.

10. Line 262-272: The role of “exit structure” by negatively charged residues was described, while this is not direct evidence for release of the product from “exit”. The authors are encouraged to show the enzyme activity of the mutants with acidic residues (Asp and Glu) to basic and acidic (Glu and Asp) residues.

Our response:

Thanks for your good suggestion. We have made a series of mutants by replacing the acidic residues Asp⁷⁰, Asp²⁸¹ and Asp³³⁵ in the exit with residue His or Asn, respectively, and the enzyme activities of the mutants were affected to varying degrees (supplementary Fig. S14a). In which, the enzyme activities of mutants BlexoHep-D70N, BlexoHep-D281N, BlexoHep-D281H and BlexoHep-D335N are extremely lower compared with that of the WT-BlexoHep, confirming that the negatively charged residues in the exit play vital roles in releasing the product from “exit”

11. Line 271: “Fig S5a” should be changed to “Fig S10a”.

Our response:

Thanks a lot. We have corrected this mistake.

12. Line 324: (I, II and II) to (I, II and III).

Our response:

Thanks a lot. We have corrected it.

13. Lines 363-373: The authors mentioned that L-shaped tunnel include the entrance by positively charged residues and the exit by negatively charged residues. In relation to 10), replacement of acidic residues with basic residues in the exit structure may provide useful

information on the role of the exit. The positively charged exit region of the mutants would exhibit the affinity with the product, indicating the role of the exit as pathway for release of the product.

Our response:

Thanks for your good suggestions. Based on your suggestion, we have mutated three acidic residues Asp⁷⁰, Asp²⁸¹ and Asp³³⁵ together in the exit tunnel to the basic residue His or Asn, which converted the electrical property of the exit tunnel from negative to positive or uncharged, and the corresponding mutants BlexoHep-D70H-D281H-D335H and BlexoHep-D70N-D281N-D335N both show too low enzyme activities to be accurately measured (supplementary Fig. S14a), which provide biochemical evidence to the role of the exit as pathway for release of the product, as you kindly suggested.

14. Line 436: sequence to sequences.

Our response:

Thanks. We have corrected it.

15. Lines 511-512: The authors should indicate molar extinction coefficient.

Our response:

Thanks. We have added the molar extinction coefficient.

16. Line 542: D2O “2” subscript.

Our response:

Thanks. We have corrected it.

17. Line 633: Remove 2003?

Our response:

Thanks for your concern. The “2003” are part of the title and thus cannot be deleted.

18. Lines 683-684: Insert space in “catalyticmechanism” and “crystalstructure”.

Our response:

Thanks a lot. We have inserted the space.

19. Line 687: Remove dot after Protein.

Our response:

Thanks. We have removed the dot.

20. Line 722: delta OK? Or Δ.

Our response:

Thanks for your concern. The authors used “delta” in the title and thus we cannot change it to “Δ”.

21. Figure1: What is small peak “Di”? The authors should isolate the product and determine its molecular weight.

Our response:

Thanks. The “Di” is an abbreviation for HP disaccharides that can be determined by comparing with the elution position of HP standard oligosaccharides as described corresponding figure legends. To more clearly show the structural property of disaccharides in each peaks, we have modified the description of the HP disaccharides in the figures and legends.

22. Figure S2: The authors should show the time-dependent degradation profile. What is the peak at retention time of around 18 min?

Our response:

Thanks for your suggestion and concern. We have shown the time-dependent degradation profile of the enzymes, as shown in the new Supplementary Fig. S4. The peak at retention time of around 18 min in supplementary Figure S2 represents polysaccharide/large oligosaccharide and enzyme protein that eluted in the void volume, which is often detected in the size exclusion chromatogram, and we specified the peaks in supplementary Fig. S2 as suggested.

Reviewer # 3 (the main findings of the study):

The authors discovered a novel Hepase with exolytic activity, exoHep. This exoHep showed high activity against hepase III resistant heparin fractions, relatively low activity against the heparin and extremely weak activities towards HS. The authors confirmed the exolytic cleavage and substrate specificity using hepase III resistant heparin and tetrasaccharide. This enzyme degrades highly sulfated HP from the reducing end. This is very unusual as nearly all enzymes acting exolytically on carbohydrates act from the non-reducing end.

The authors elucidated the protein structure and protein-heparin interactions through

crystallization of SeMet labeling followed by single wavelength anomalous dispersion phasing.

Finally, the authors used this exoHep successfully sequenced the 5 octasaccharide. They proposed the mechanism of the exolytic mode of exoHep and strategy for exo-sequencing using exoHep.

This new exoHep provides a new tool for HP/HS sequencing.

The manuscript is well written.

Reviewer #3 (Revisions needed):

(Comments)

1. The most unusual feature of these exo enzymes is that they act from the reducing end. While the author's data suggest this is the case they should apply a second method to confirm. The author's should test whether it is possible to disrupt enzyme action by first reducing the hemiacetal at the oligosaccharides reducing end or by reductively aminating the reducing end with 2-aminoacridone. This would help confirm that action does take place from the reducing end.

Our response:

Thanks for your good suggestion. Based on your suggestion, we labeled the reducing end of the HP DP13. As the results shown in Supplementary Fig. S5, the 2-AB-labeling at the reducing ends of HP DP13 completely inhibited the action of the PL15_2 Hepases compared with the quick degradation of unlabeled HP DP13 by these enzymes (Fig. 1a-e), indicating that the introduction of 2-AB group at the reducing end of HP chain hindered the action of the PL15_2 Hepases and further confirmed that these enzymes are exolytic Hepases that act from the reducing ends of substrates.

2. The authors should also use RI detector instead of UV 232 nm used in Fig 1? This way they can see both oligosaccharide and disaccharide changing at the same time.

Our response:

Thanks for your concern and suggestion. As you suggested, we can detect both oligosaccharide and disaccharide changing by using RI detector, but this way will make the results very confusion due to the appearance of many peaks corresponding to

oligosaccharides with different size, which will seriously disrupt the judgment of the exo- or endolytic characters of the enzymes. In fact, compared with RI detector the biggest advantage of UV232 nm is the ability to distinguish the unsaturated saccharides produced by enzyme from the saturated substrates such as the saturated HP DP13 or polysaccharide, otherwise we cannot judge the peaks from the reducing ends or nonreducing ends of the test substrates. Taken together, the application of UV232 nm can make the judgment of substrate-degradation mode of the enzymes much easier, and the only disadvantage is difficult to detect the changing of the saturated substrates as you concerned. To show the presence of substrate in Fig. 1, we have showed the SEC chromatogram of the saturated HP DP13 detected at 210 nm that is much more sensitive than RI detector and can save precious sample a lot.

3. What's the purity of the five octasaccharides? Did author confirm the octasaccharide chain using another sequencing method for comparison to the exoHep sequencing method?

Our response:

Thanks for your concerns. As shown in the manuscript, the purity and compositions of the five octasaccharides were verified by HILIC-ESI-MS (Supplementary Fig. S17). Based on the results, except for P8-3 with some minor component other four octasaccharides are relative pure. Furthermore, based on the suggestions from you and another reviewer the sequences of the five octasaccharides were confirmed by using the MS/MS analysis as shown in the new Supplementary Fig S18, Fig S19, Fig S20, Fig S21 and Fig S22.

4. Figure 5, the color of exoHep is too light. It is suggested to use a darker background.

Our response:

Thanks a lot. We have changed the color as you suggested.

REVIEWERS' COMMENTS

Reviewer #1 (Remarks to the Author):

The authors have provided an extensive amount of high quality additional data. This is a very nice piece of work, which is of high interest to the field. I ask for one further edit to be made before publication, which I have detailed below:

Line 173 – Please change “obviously” to obvious.

Line 377 – Please edit the text to remove the UA-GlcNAc3S, as there is also a possibility that the fragment is a result of two cleavages from another fragment. It is very challenging to sequencing this monosaccharide. I would also ask to please edit the table to

P8-3 $\Delta^?$ -4HexA2S1-4GlcNS6S-4HexA2S1-4GlcNS6S1-4HexA1-4GlcNS6S

P8-5 $\Delta^?$ -4HexA2S1-4GlcNS6S1-4HexA2S1-4GlcNS6S1-4HexA2S1-4GlcNS6S

And edit the text to:

To verify the enzymatic sequencing, the compositions and sequences of these octasaccharides were subjected to ESI-MS (Supplementary Fig. S17 17 and Table. S7) and ESI-MS/MS (Supplementary Fig. S18, Fig. S19, Fig. S20, Fig. S21 and Fig. S22). Notably, the monosulfated disaccharide located at the nonreducing end of P8-3 or P8-5 could not be deduced by the enzymatic method because of lack of it's corresponding disaccharide standard.

Or remove P8-3 and P8-5 from the manuscript as there are three other sequenced structures to validate the point.

Also, for the figures S18-S22, I could not see the structure in pdf, but this may be a download issue.

Thank you, Rebecca Miller

Reviewer #2 (Remarks to the Author):

The manuscript has been greatly improved by the revision. Especially, the authors have freshly obtained the mutant enzyme (BlexoHep-D70H-D281N-D335H) showing a certain endolytic activity through the molecular conversion of mode of action, and suggested structural determinants for the exolytic activity of PL15_2 family enzymes. The revised manuscript describes clearly the functional and structural aspects of novel heparin lyases in PL15_2 family and gives an insight into the mode of action of the exolytic enzyme. The reviewer thinks this manuscript will be acceptable for publication in Nature Communications.

Reviewer #3 (Remarks to the Author):

The authors have adequately addressed all of the reviewer's comments.

Our responses to the reviewers' comments have been described below one by one referring to the pages and locations of the marked manuscript in yellow.

Reviewer # 1 (the main findings of the study):

The authors have provided an extensive amount of high quality additional data. This is a very nice piece of work, which is of high interest to the field. I ask for one further edit to be made before publication, which I have uploaded in a word document.

Thank you, Rebecca Miller

Reviewer #1 (Revisions needed):

(Comments)

1. Line 173 – Please change “obviously” to obvious.

Our response:

Thanks a lot. We have changed as suggested.

2. Line 377 – Please edit the text to remove the UA-GlcNAc3S, as there is also a possibility that the fragment is a result of two cleavages from another fragment. It is very challenging to sequencing this monosaccharide. I would also ask to please edit the table to

P8-3 $\Delta^?$ -4HexA2S1-4GlcNS6S-4HexA2S1-4GlcNS6S1-4HexA1-4GlcNS6S

P8-5 $\Delta^?$ -4HexA2S1-4GlcNS6S1-4HexA2S1-4GlcNS6S1-4HexA2S1-4GlcNS6S

And edit the text to:

To verify the enzymatic sequencing, the compositions and sequences of these octasaccharides were subjected to ESI-MS (Supplementary Fig. S17 17 and Table. S7) and ESI-MS/MS (Supplementary Fig. S18, Fig. S19, Fig. S20, Fig. S21 and Fig. S22). Notably, the monosulfated disaccharide located at the nonreducing end of P8-3 or P8-5 could not be deduced by the enzymatic method because of lack of it's corresponding disaccharide standard.

Or remove P8-3 and P8-5 from the manuscript as there are three other sequenced structures to validate the point.

Also, for the figures S18-S22, I could not see the structure in pdf, but this may be a download issue.

Our response:

Thanks a lot. We have edited the manuscript as suggested. We edit the text to:

To verify the enzymatic sequencing, the compositions and sequences of these octasaccharides were subjected to ESI-MS (Supplementary Fig. S17 17 and Table. S7) and ESI-MS/MS (Supplementary Fig. S18, Fig. S19, Fig. S20, Fig. S21 and Fig. S22). Notably, the monosulfated disaccharide located at the nonreducing end of P8-3 or P8-5 could not be deduced by the enzymatic method because of lack of it's corresponding disaccharide standard.

Reviewer # 2 (the main findings of the study):

The manuscript has been greatly improved by the revision. Especially, the authors have freshly obtained the mutant enzyme (BlexoHep-D70H-D281N-D335H) showing a certain endolytic activity through the molecular conversion of mode of action, and suggested structural determinants for the exolytic activity of PL15_2 family enzymes. The revised manuscript describes clearly the functional and structural aspects of novel heparin lyases in PL15_2 family and gives an insight into the mode of action of the exolytic enzyme. The reviewer thinks this manuscript will be acceptable for publication in Nature Communications.

Our response:

Thanks for your kindly comments.

Reviewer # 3 (the main findings of the study):

The authors have adequately addressed all of the reviewer's comments.

Our response:

Thanks for your kindly comments.